# The Structural Integrity of the Model Lipid Membrane during Induced Lipid Peroxidation: The Role of Flavonols in the Inhibition of Lipid Peroxidation

**DOI:** 10.3390/antiox9050430

**Published:** 2020-05-15

**Authors:** Anja Sadžak, Janez Mravljak, Nadica Maltar-Strmečki, Zoran Arsov, Goran Baranović, Ina Erceg, Manfred Kriechbaum, Vida Strasser, Jan Přibyl, Suzana Šegota

**Affiliations:** 1Ruđer Bošković Institute, 10000 Zagreb, Croatia; Anja.Sadzak@irb.hr (A.S.); nstrm@irb.hr (N.M.-S.); Goran.Baranovic@irb.hr (G.B.); ierceg@irb.hr (I.E.); Vida.Strasser@irb.hr (V.S.); 2Faculty of Pharmacy, University of Ljubljana, 1000 Ljubljana, Slovenia; janez.mravljak@ffa.uni-lj.si; 3Jožef Stefan Institute, 1000 Ljubljana, Slovenia; zoran.arsov@ijs.si; 4Institute of Inorganic Chemistry, Graz University of Technology, 8010 Graz, Austria; manfred.kriechbaum@tugraz.at; 5CEITEC, Masaryk University, 62500 Brno, Czech Republic; jan.pribyl@ceitec.muni.cz

**Keywords:** bilayer thickness, elasticity, flavonols, fluidity, lipid peroxidation, myricetin, myricitrin, quercetin

## Abstract

The structural integrity, elasticity, and fluidity of lipid membranes are critical for cellular activities such as communication between cells, exocytosis, and endocytosis. Unsaturated lipids, the main components of biological membranes, are particularly susceptible to the oxidative attack of reactive oxygen species. The peroxidation of unsaturated lipids, in our case 1,2-dioleoyl-sn-glycero-3-phosphocholine (DOPC), induces the structural reorganization of the membrane. We have employed a multi-technique approach to analyze typical properties of lipid bilayers, i.e., roughness, thickness, elasticity, and fluidity. We compared the alteration of the membrane properties upon initiated lipid peroxidation and examined the ability of flavonols, namely quercetin (QUE), myricetin (MCE), and myricitrin (MCI) at different molar fractions, to inhibit this change. Using Mass Spectrometry (MS) and Fourier Transform Infrared Spectroscopy (FTIR), we identified various carbonyl products and examined the extent of the reaction. From Atomic Force Microscopy (AFM), Force Spectroscopy (FS), Small Angle X-Ray Scattering (SAXS), and Electron Paramagnetic Resonance (EPR) experiments, we concluded that the membranes with inserted flavonols exhibit resistance against the structural changes induced by the oxidative attack, which is a finding with multiple biological implications. Our approach reveals the interplay between the flavonol molecular structure and the crucial membrane properties under oxidative attack and provides insight into the pathophysiology of cellular oxidative injury.

## 1. Introduction

Lipid peroxidation is a complex process associated with the oxidative deterioration of lipids and the production of various breakdown products [1,2]. Lipid hydroperoxides and conjugated dienes or trienes are considered primary oxidation products, which, due to their instability, break down and form secondary oxidation products, among which are aldehydes, ketones, hydrocarbons, alcohols, and others [3]. Lipid peroxidation occurrence in the human body is a consequence of oxidative stress, which has been correlated with various diseases and health conditions, including neurodegenerative diseases, heart and cardiovascular system conditions, inflammatory immune injuries, and others [4]. The presence of polyunsaturated fatty acids (PUFAs) in a phospholipid bilayer makes it highly susceptible to oxidative damage, resulting in changes in the membrane properties. For example, a decrease in the fluidity leads to the loss of its functionality as a barrier. Furthermore, it has been suggested that some peroxidation products, in particular malondialdehyde (MDA), 4-hydroxy-2,3-nonenal (HNE), and other 4-hydroxy-2,3-alkenals (HAKs) of different chain lengths can affect several cell functions, including signal transduction, gene expression, cell proliferation, and the response of the target cell [5]. For example, Cajone and Bernelli-Zazzera [6] showed that HNE caused an increase in the expression of the hsp 70 gene in human hepatoma cells. Moreover, it was shown that HNE can cause the activation of heat shock factor in vitro [7]. To maintain homeostasis, it is crucial to achieve balance between a steady formation of pro-oxidants and a similar rate of their consumption by antioxidants. If the continuous regeneration of antioxidants is not sufficient, oxidative damage occurs, resulting in pathophysiological events [8,9]. 

Flavonoids are natural antioxidants with the ability to act as reducing agents, hydrogen donors, and singlet oxygen quenchers [10]. Additionally, they possess a metal-chelating potential [11,12]. They are polyphenolic compounds, consisting of fifteen carbon atoms arranged in a C6-C3-C6 configuration. Two aromatic rings are connected through a 3-carbon bridge, usually in the form of a heterocyclic ring [13]. The structural characteristics that are presumably most important for their antioxidant activity are a hydroxylated B-ring (Figure 1) and the presence of the double bonds C2=C3 and C=O in the C-ring. 

Depending on their substituents, flavonoids can further be classified into several subcategories: flavonols, flavones, catechins, flavanones, anthocyanidins, and isoflavonols [14]. Various antioxidants, including polyphenols, are commonly found in foods and beverages of plant origin, including fruits, vegetables, cocoa, tea, and wine. Therefore, they play an essential role in promoting preventive health care through diet [13,15]. The antioxidant activity of a polyphenolic compound is determined by the ability of the phenolic hydrogen to scavenge free radicals [16]. Although almost all subcategories of flavonoids exhibit antioxidative activity, it has been reported that flavones and catechins demonstrate the most powerful protection against reactive oxygen species [15]. In relation to their free radical scavenging and metal ions-chelating activities, a multitude of health-promoting effects have been observed, including anti-inflammatory, anti-mutagenic, and anti-carcinogenic. Furthermore, they are studied as key cellular enzyme functions modulators [15].

Flavonol protection against oxidative stress can be achieved through different mechanisms, with the direct scavenging of free radicals being one of them. Highly reactive hydroxy groups can react with the radical and stabilize the reactive oxygen species, which is followed by the inactivation of the radical [15]. Apart from direct scavenging, flavonoids can inhibit lipid peroxidation by the chelation of metal ions which usually catalyze the reaction. The impact of transition metal ions on lipid peroxidation has been extensively studied [17,18,19,20,21]. Fenton and Fenton-like reactions are often used to explain the production of hydroxyl radicals, since it was demonstrated that even trace levels of cellular transition metal ions can catalyze a Fenton reaction in vivo at the physiological level of hydrogen peroxide [18,22].

The interaction of polyphenolic compounds and lipid membranes could be one of the mechanisms in the protection against peroxidation. Depending on the lipid and flavonol chemical structure and composition, the interaction between flavonols and biological or model membranes can result in the binding of flavonol at the lipid–water interface or the distribution of flavonol in the hydrophobic part of the bilayer [23]. For example, Van Dijk et al. [24] investigated the relationship between the relative hydrophobicity of flavonoids and their binding with the liposomes using fluorescence quenching. The affinity of flavonoids from the subcategory of flavonols for liposomes was determined to be much higher than the one of flavanones, which was explained by different structural characteristics. It was determined that the planar configuration of flavonols, in contrast to the tilted configuration of flavanones, favors intercalation into vesicle membranes. If the flavonols partition in the non-polar region of the bilayer, they can influence the membrane fluidity. If the membrane is rigid, the probability of lipid radical interactions with other fatty acids is increased, due to the limited motion of fatty acid chains [25]. The more hydrophilic flavonols can form hydrogen bonds with the polar head groups at the lipid–water interface of the membrane and provide a level of protection for the bilayer. Furthermore, different subgroups of flavonols can have different effects on the phase transition temperature of various lipids, making membranes more or less ordered [23].

Within this study, we used three different flavonols: quercetin (QUE), myricetin (MCE), and myricitrin (MCI) (Figure 1), which are all found in various plants. For example, QUE can be found in onions, apples, and berries. MCE is also found in various berries and vegetables, as well as in teas and wines produced from various plants [26]. MCI has been extracted from numerous plants, such as *Manilkara zapota* and *Eugenia uniflora* [27]. QUE is one of the most abundant and most studied flavonols, owing to its antioxidant and anti-inflammatory properties [28,29]. Furthermore, it has been shown that QUE exhibits a hepatoprotective, antifibrotic [30], anti-coagulative [31], and antimicrobial [32] activity. MCE has, among others, been suggested as a good candidate for the development of new drugs for the treatment of Alzheimer’s disease due to its strong free radical-scavenging activity, which can block Aβ-induced neuronal death [28]. Its rhamnose glycoside, MCI, shows numerous beneficial effects as well, including anti-allergic [33], antioxidant, anti-inflammatory, antifibrotic [34], and antinociceptive activity [35]. The three flavonols differ in substituents and, consequently, hydrophobicity. 

To investigate the protective effect of three structurally different flavonols on the molecular structure, integrity, and elasticity of lipid membranes under oxidative attack, we used a combination of techniques. The membrane integrity was measured by its crucial structural parameters, such as elasticity, surface roughness, thickness, and fluidity. We hypothesized that the insertion of antioxidative flavonols would suppress the disintegration of the lipid membranes during the lipid peroxidation. Three different molar fractions (*x* = 0.01, 0.05, 0.1) of flavonols were chosen. Additionally, we wanted to highlight the dependence of the preservation of the membrane on the flavonol hydrophobicity, i.e., on their localization inside the bilayer. The most hydrophobic flavonol was expected to be hosted deep within the membrane and vice versa. As the model membrane, 1,2-dioleoyl-sn-glycero-3-phosphocholine (DOPC) was chosen, which possesses two chains of monounsaturated fatty acids. Due to the absence of adjacent double bonds with methylene bridges between them, which occur in PUFAs, the initiation phase of the lipid peroxidation is slower, while the overall mechanism is essentially the same in mono- and polyunsaturated fatty acids [36]. The main difference is in the occurrence of different peroxidation products, which are formed as a consequence of conjugation. Mass Spectrometry (MS) was used to identify and compare the lipid peroxidation products before and after the incorporation of flavonols in different molar fractions, while Fourier Transform Infrared Spectroscopy, Attenuated Total Reflectance technique, (FTIR-ATR) was used to determine the extent of the lipid peroxidation reaction and quantify the inhibition in the samples with flavonols. Atomic Force Microscopy (AFM) and Force Spectroscopy (FS) were used to analyze the nanomechanical and topological properties of the lipid bilayers, such as roughness (determined from the AFM), thickness, and elasticity defined via Young’s modulus (determined from force–distance curves (FS)). Dynamic Light Scattering and electrophoretic measurements (DLS/ELS), as well as Electron Paramagnetic Resonance (EPR) and Small Angle X-Ray Scattering (SAXS) measurements provided further information on the flavonol antioxidative role in the preservation of membrane integrity. Our results demonstrate the unique capability of this multi-technique approach and indicate its potential to deeply enhance the understanding of cellular oxidative injury.

## 2. Materials and Methods

### 2.1. Chemicals

Iron(II) chloride tetrahydrate (98%) was purchased from Alfa Aesar, (Ottawa, ON, Canada). Myricetin (>97%) and myricitrin (>98%) were purchased from TCI Chemicals Pvt. Ltd. (Chennai, India). Quercetin (≥95%) and phosphate-buffered saline (PBS) buffer (PBS tablets, pH 7.4, *I_c_* = 150 mM) were purchased from Sigma-Aldrich (St. Louis, MO, USA). 1-2-dioleoyl-sn-glycero-3-phosphocholine (DOPC) was supplied by Avanti Lipids (Industrial Park Drive Alabaster, AL, USA). Coumarin hydrazine (CHH) was synthesized at the UL-Faculty of Pharmacy. Methanol (99.5%) and hydrogen peroxide (30% p.a.) were purchased from Kemika, Zagreb. Chloroform (99.93%) p.a., was purchased from Lach-ner Ltd. (Neratovice, Czech). The EPR probe 5-DOXYL-stearic acid and 2,2-diphenyl-1-picrylhydrazyl (DPPH) were purchased from Sigma Aldrich (St. Louis, MO, USA). All the chemicals were used without further purification.

### 2.2. Preparation and Oxidation of Liposomes with and without Inserted Flavonols

The DOPC liposomes were prepared by dissolving DOPC in chloroform. The QUE, MCE, and MCI were inserted in three different molar fractions (0.01, 0.05, 0.1) by mixing methanol solution of each flavonol with chloroform solution of DOPC. Solvents were evaporated using a rotary evaporator and the remaining films were dried in a vacuum. The dried films were dispersed in PBS by manual shaking at room temperature. During rehydration, the lipid film was gradually scraped off the wall of the glass bottle by alternately immersing the flask in the ice and hot water. The liposome suspension was left to stabilize overnight. Multilamellar liposomes were used to avoid the loss of lipids and flavonols during the process of extrusion, except for the DLS measurements. The concentration of lipids was adjusted for different methods and will be stated in the corresponding sections. The lipid peroxidation was initiated by the addition of FeCl_2_ × 4H_2_O and H_2_O_2_. The final concentrations of FeCl_2_ × 4H_2_O and H_2_O_2_ in the suspension were 1 mM, and the reaction was advanced for 1 h before measurements.

Hydrogen peroxide emerged as a major redox metabolite operative in redox sensing, signaling, and regulation. The concentration of extracellular H_2_O_2_ in redox signaling under physiological conditions was between 0.1 μM and 10 μM in oxidative eustress conditions. Higher extracellular concentrations (5–500 μM) led to adaptive stress responses. Supraphysiological extracellular concentrations in oxidative distress (5 μM < *c* < 1 mM) led to the irreparable damage of biomolecules [37]. Since we wanted to initiate damage of the lipid molecules, we used the 1 mM of H_2_O_2_ as it was in the extracellular solution under oxidative distress conditions.

In our experiments, the lipid peroxidation process was initiated by the addition of hydrogen peroxide and iron(II) ions (Fenton reaction) to the liposome suspension, where hydroxyl radicals were formed mainly by one-electron redox reactions between the H_2_O_2_ and the pre-existing hydroperoxides with transition metal ions. The reactions of ferrous (Fe^2+^) ions with hydrogen peroxide and oxygen can generate ferryl and perferryl species, which are strong oxidants and have also been suggested to be capable of initiating radical reactions. In relation to the present approach, it must be mentioned that there are other physiological conditions that induce lipid peroxidation, for example systems including ascorbate and Fe^2+^ ions. Even vitamin C, which is a known antioxidant, can act as a prooxidant. Under favourable conditions, it contributes to the oxidative damage of lipids by reducing Fe^3+^ to Fe^2+^ ions (or Cu^2+^ to Cu^+^) [38]. The ascorbate/Fe^2+^ system was used in a study of peroxidation and was found to cause a significant degradation of ethanolamine phosphoglycerides [39]. Moreover, it has been suggested that the prooxidant effect of vitamin C could not have relevance in vivo [40]. Therefore, by performing experiments using a H_2_O_2_/Fe^2+^ system to initiate lipid peroxidation, two opposite (antioxidant and prooxidant) effects were avoided.

### 2.3. EPR Spectroscopy

The EPR spectra were collected by a home-modified Varian E-109 spectrometer (Ruđer Bošković Institute, Zagreb, Croatia) using a Bruker ER 041 XG microwave bridge working at a microwave frequency of 9.3 GHz (i.e., X-band) at room temperature (25 °C). The temperature in the EPR cavity was controlled by a Bruker ER 4111 temperature controller using a nitrogen gas flow accurate to 1 °C. 

#### 2.3.1. Antioxidant Activity

For the determination of the antioxidant activity, the spectrometry settings were: magnetic field modulation frequency 100 kHz, central field 331.0 mT, sweep range 10 mT, sweep time 20 s, microwave power 10 mW, and modulation amplitude 0.1 mT. A 2,2-diphenyl-1-picrylhydrazyl (DPPH) stable free radical was used to monitor the scavenging capability of flavonols using EPR spectroscopy. The stability of the freshly prepared ethanol solution was checked, and no significant loss of signal in the EPR signal was recorded within 24 h. A 4050 μL volume of a DPPH stock ethanol solution (0.5 mM) was added to 450 μL of a flavonol solution and mixed. The final solution was promptly inserted into the EPR capillary, which was then placed in a standard quartz tube. The EPR spectra were collected as a function of time initiated by contact with the sample and radical solution. The scavenging effect of the flavonol samples on DPPH radicals was obtained from the EPR signal intensities calculated by the double integration of the EPR spectra and expressed in arbitrary units. The signal intensity of the pure 0.5 mM DPPH solution in PBS, recorded just before starting the sample evaluation, was set as the reference signal intensity (*I*_0_) for the reaction time *t* = 0 min. The EPR signal intensity of DPPH radicals was decreased upon the flavonol addition and monitored for 20 min in recording intervals of 0.5 min and 1.0 min, depending on the sample activity. The remaining signal intensity, i.e., the remaining DPPH radicals after the reaction time, t, normalized and expressed as a percentage, *I*_N_, was calculated as: *I*_N_ = (*I*/*I*_0_) × 100, where *I* is the signal intensity of the DPPH radicals in the flavonol solution measured at time, *t*. Each sample was analyzed in triplicate. The results are presented as mean values.

#### 2.3.2. Fluidity Change during Lipid Peroxidation

The experimental parameters for monitoring the fluidity change upon the initiated lipid peroxidation were: central magnetic field 331.0 mT, sweep width 10 mT, modulation amplitude 0.1 mT, and microwave power 4.9 mW. A standard Bruker ER 4111 VT temperature controller with a nitrogen gas flow was used to control the temperature within 1 °C. A manganese standard reference, Mn^2+^ in MnO, was used to calibrate the magnetic field of the EPR spectrometer. The EPR spectra were simulated with a custom-built program in MATLAB (The MathWorks Inc., Natick, MA, USA) using the EasySpin program package (Stoll and Schweiger, 2006) to extract the spectral parameters—either one component with a slow dynamic or two components with different dynamics. A three-line, narrow EPR spectra is typical for nitroxide free radicals undergoing rapid isotropic motion, which can be characterized with *a*_oN_. The value of *a*_oN_ was taken to be one half of the difference in the resonance fields of the high- and low-field lines. For the slow component, the distance between the outer peaks (2*A*_zz_) was monitored. 

All the samples for EPR spectroscopy were prepared with spin probe 5-doxyl stearic acid (5-DSA) dissolved in ethanol (1% v/v of 200 mM).

### 2.4. High Resolution Mass Spectrometry

#### 2.4.1. Derivatization with 7-(Diethylamino)coumarin-3-carbohydrazide (CHH)

Due to the low concentrations of lipid peroxidation products (LPPs) as well as the low ionization efficiencies—i.e., their low proton affinities—their chemical derivatization is necessary. It provides high proton affinities and enhances their ionization in positive ion mode and thus provides the detection of numerous LPPs. We used CHH for derivatization, assuming a rapid and specific reaction between CHH and aldehydes and ketones. Additionally, the hydrophobicity and relatively high mass of CHH were expected to enable the simultaneous extraction of both short aliphatic and nonpolar high molecular weight carbonyls with organic solvents. The CHH derivatization enhanced the ionization of both aliphatic and lipid-bound carbonyl-containing LPPs, giving access to both small, aliphatic, and water-soluble and large, nonpolar, lipid-esterified carbonylated species. The oxidized liposomes (1.5 mM) were individually derivatized with CHH (50 μL, 10 mM) at 37 °C for 1 h (Figure 2). After derivatization, the samples were extracted in the same volume of chloroform, diluted with a mixture of methanol and chloroform (2:1, v/v), and analysed immediately.

#### 2.4.2. Measurement Parameters

The samples were diluted (to 10 pmol/μL) in a mixture of methanol and chloroform (2:1, v/v) and analyzed by direct infusion using a Q Exactive™ Plus Hybrid Quadrupole-Orbitrap™ Mass Spectrometer (Thermo Scientific™). Type of mass detector: Orbitrap measuring range *m*/*z*: 50–6000 *m*/*z*; mass resolution: 140,000 FWHM (full width at half maximum); mass accuracy: <1 ppm with internal calibration, <3 ppm with external calibration. The MS spectra were acquired in Fourier Transform Mass Spectrometry (FT-MS) scan mode with a target mass resolution of 100,000 at *m*/*z* 400. The acquisition period was 15 min. The recorded spectra were analyzed with a Thermo Xcalibur Qual Browser (Xcalibur 4.2 SP1, Thermo Fisher Scientific Inc., Waltham, MA, USA). All the spectra were manually searched throughout the whole timeframe for all the suspected aliphatic carbonyl compounds with a mass accuracy of 5 ppm.

### 2.5. FTIR-ATR Spectroscopy

The concentration of DOPC for the FTIR measurements was adjusted to 20 mg mL^−1^. The spectroscopic measurements were performed using a PerkinElmer “Spectrum 400 Series” spectrometer (Jožef Stefan Institute, Ljubljana, Slovenia) equipped with a Horizon ATR accessory (Harrick Scientific) with a trapezoidal germanium crystal. Each spectrum was collected at a nominal resolution of 4 cm^−1^ resolution and as mean value of 32 spectra. A special holder for the ATR crystal was used. It was placed in contact with an aluminum block and the temperature was controlled by a circulating water bath. All the spectra were collected at 40 °C. A quantity of 200 μL of each sample was placed onto the ATR crystal and spread over the whole area. The sample was dried by purging with dry nitrogen until there was no significant change in the broad band at 3200–3600 cm^−1^, which corresponded to ν(O–H) of the solvent [41].

#### Data Analysis—The Extent of Lipid Peroxidation

A data analysis was performed by modifying the procedure used by Oleszko et al. [42]. Since lipid peroxidation should lead to a change in the integral absorbance of the ν(C=O) band at 1737 cm^−1^, it was analyzed with respect to the ν_as_(CH_3_) at 2959 cm^−1^ band, where we expect no changes in the absorbance after the reaction [42]. The values of the integral absorbances of both bands were calculated using an Origin Pro 9. According to Bradley and Kretch [43], the IR spectra of solids usually consist of peaks which can be described using a Gaussian function, while gases are dominantly fitted with a Lorentzian function. Since the lipid membranes were in the fluid phase, the peak shape was expected to be a combination of Gaussian and Lorentzian curves. When fitted with a linear combination of Gaussian and Lorentzian curves, the bands in the interval 2800–3100 cm^−1^ showed a large Lorentzian character, while the band in the interval 1600–1800 cm^−1^ showed a large Gaussian character. This can be explained by the fact that this peak is composed of several slightly shifted Lorentzian peaks belonging to different species formed during the lipid peroxidation, which cannot be resolved. Therefore, peaks in the interval 2800–3100 cm^−1^ were fitted to a pure Lorentzian curve, while the peak in the interval 1600–1800 cm^−1^ was fitted to a pure Gaussian curve. In the first iteration, all the peak parameters were included in the fit. The obtained band widths were then used to compute the average band widths for the systems containing the selected flavonol. Finally, the integral absorbances were recomputed using fixed peak widths in order to reduce the parameter dependencies.

The ratio of the integrated absorbances *A_i_* of the *i*-th sample (*i* = 0 without flavonols, *i* = 1, 2, 3 with flavonols) was calculated according to:(1)$$r_{i}=\frac{A_{i}\left(\nu \left(C=O\right)\right)}{A_{i}\left(\nu_{\mathrm{as}}\left({\mathrm{CH}}_{3}\right)\right)},$$
for all the samples before and after lipid peroxidation. The change in that ratio after lipid peroxidation (LP), $r_{i}^{\mathrm{LP}}$ was determined relatively to that of the control sample, which is the liposome suspension before the occurrence of the reaction:(2)$$\rho_{i}=\frac{r_{i}^{\mathrm{LP}}-r_{i}}{r_{i}}.$$

If the number of C=O bonds increases, the ratio $\rho_{i}$ increases. Finally, the inhibition for each flavonol and each molar ratio was calculated using the formula:(3)$$R_{i}=\frac{\rho_{i}-\rho_{0}}{\rho_{0}},$$
where *ρ*_0_ corresponds to the relative ratio of the integrated absorbances for the case of DOPC without inserted flavonols (*i* = 0). If added flavonols hinder the lipid peroxidation, *R_i_* exhibits negative values with a minimum value of −1, corresponding to the total inhibition of the reaction. In that case, the value *R_i_* is a measure of the inhibition of the lipid peroxidation reaction (Appendix B, Appendix A).

### 2.6. Dynamic Light Scattering (DLS) and Electrophoretic (ELS) Measurement

A photon correlation spectrophotometer equipped with a 532 nm green laser (Zetasizer Nano ZS, Malvern Instruments, Worcestershire, UK) was used for the determination of the size distribution and zeta potential of the unilamellar DOPC liposomes (Avanti’s Mini-Extruder with 100 nm membrane) in PBS at (25.0 ± 0.1) °C. The final concentration of the suspension was 0.2 mg mL^−1^. The intensity of the scattered light was measured at a 173° angle. The hydrodynamic diameter (*d*_H_) was determined from the peak maximum of the volume size function. The zeta potential (*ζ*) was calculated from the electrophoretic mobility using a Smoluchowski approximation (f(*κa*) = 1.5). The hydrodynamic radius values were reported as an average value of 10 measurements, while the zeta potential values were reported as an average of 3 independent measurements. The data processing was carried out using Zetasizer Software 7.13 (Malvern Instruments LTD, Malvern, Worcestershire, UK).

### 2.7. Small Angle X-ray Scattering (SAXS)

For the SAXS measurements, dried films were dispersed in water instead of PBS to avoid a decrease in the electronic contrast and final signal that could appear using the PBS buffer solution. To initiate lipid peroxidation, the dried films were resuspended in the water solutions of FeCl_2_ × 4H_2_O (10 mM, 250 μL) and H_2_O_2_ (10 mM, 250 μL). The lipid peroxidation reaction advanced for 1 h before recording. The final concentration of the samples for the SAXS measurements was 50 mg mL^−1^.

The SAXS measurements were carried out in transmission mode at defined temperatures by a laboratory SAXS instrument (SAXS-Point 2.0, Anton Paar, Graz, Austria). The SAXS camera was equipped with a micro-X-ray source operating at 50 W (point-focus) using Cu-K_α_-radiation (*λ* = 0.1542 nm) and a 2D X-ray detector (EIGER2 R 500K, Dectris, Switzerland). The SAXS patterns were recorded at 571 mm sample-to-detector distance. All the isotropic 2-dimensional SAXS patterns were azimuthally averaged to 1-dimensional SAXS-curves. The SAXS curves of pure water were taken for background subtraction. The angular *q*-range was 0.01 nm^−1^ to 6 nm^−1^, with q being the magnitude of the scattering vector, which corresponds to a total 2*θ* region of 0.14° to 7° applying the conversion *q* [nm^−1^] to 2*θ*(°) using Equation (5). The sample cell in the X-ray beam was a quartz capillary (1 mm diameter, wall thickness of 10 µm) with two vacuum tight screwcaps on both ends inserted into a thermostatted sample-stage set to a defined temperature (30 °C). The vacuum in the camera during the measurement was kept at ≈1 mbar. The exposure time was 300 s times 3.

The analysis of the scattering data of liposome structures after the lipid peroxidation was performed using the programs GIFT [44] and DECON [45], developed by Otto Glatter. GIFT (Generalized Indirect Fourier Transformation) is based on the simultaneous determination of the form and the structure factor.

The scattering intensity is expressed by the following equation:(4)$$P\left(q\right)=nP\left(q\right)S\left(q\right),$$
where *P*(*q*) and *S*(*q*) are the form and structure factor, respectively, and n is the number density of the particles. *P*(*q*) describes the internal structure of the particles, while *S*(*q*) describes the interaction between the particles. The value *q* is the magnitude of the scattering vector and is related to the scattering angle by the following equation: (5)$$q=\frac{4\pi }{\lambda }\sin\left(\theta \right)\;,$$
where *λ* is the wavelength of the X-ray and 2*θ* is the angle of the scattered X-rays. For the lamellar structure, the form factor *P*(*q*) can be expressed as:(6)$$P\left(q\right)=\frac{2\pi r^{2}A}{q^{2}}P_{l}\left(q\right),$$
where *A* is the area of the lamellar phase. The relation between *P_l_*(*q*) and the normal bilayer pair distribution function *p_l_*(*r*) is the Fourier transformation shown in (7), while the relation between *P_l_*(*q*) and the self-correlation function of electron density function Δ*ρ*(*r*) is the Fourier transformation in (8):(7)$$P_{l}\left(q\right)=2\int_{0}^{\infty }p_{i}\left(r\right)\cos\left(qr\right)dr,$$
(8)$$p_{l}\left(r\right)=2\int_{0}^{\infty }\rho_{l}\left(r_{l}\right)\Delta \rho_{l}\left(r+dr\right)dr_{l}.$$

A Fourier analysis of the multilamellar SAXS patterns (with the sharp Bragg peaks) was performed using the in-house Javascript program available online [46]. The input parameters were: the number of the visible first Bragg peaks (2); the intensities of the two peaks; the lamellar *d*-spacing of the peaks (6.1 nm); and the sign of the amplitudes (square roots of the intensities) of the two peaks, which can be either + or −. Only the combination of − for the two peak amplitudes gave a reasonable result for the electron density.

### 2.8. Atomic Force Microscopy (AFM) and Force Spectroscopy (FS)

#### 2.8.1. Preparation of the Supported Lipid Bilayers (SLBs)

The procedure for the preparation of SLBs is the drop deposition method, which has already been reported [47]. Briefly, a drop of multilamellar vesicles (MLVs) suspension (100 µL) was added to the fluid cell with a freshly cleaved mica plate and thermostated at 25 °C. After 10 min incubation, due to electrostatic interactions between the DOPC liposomes and mica, the liposomes adsorbed on the mica surface and formed SLB [48]. The unadsorbed liposomes were removed by washing the surface with the filtered (0.22 µm Whatman) PBS solution. 

#### 2.8.2. AFM Imaging in Fluid and FS Measurements before and after Lipid Peroxidation

AFM images were obtained by scanning the SLBs on the mica surface in the fluid using an AFM FastScan Dimension (Bruker Billerica, USA), operated using the PeakForce QNM mode sing Scanfastsyst—Fluid + Bruker probes, with the spring constant (Nom. *k* = 0.7 Nm^−1^; Nom. resonant freq. *ν* = 150 kHz). The imaging was performed at 25.0 °C, allowing thermal equilibration before each sample imaging. The thermal tune method was used for the cantilever calibration as previously described [47,49]. AFM images were collected using random spot surface sampling (at least two areas per sample) for each analyzed sample. The quantitative mechanical data was obtained by employing DMT modulus within the Bruker software. All the images were processed by first-order two-dimensional flattening only and analyzed using the NanoScope Analysis software (Version 1.9). 

### 2.9. Statistical Data Analysis

Where applicable, the obtained data have been presented as the mean value and the standard deviation. A general linear model (GLM) for ANOVA has been used for the statistical comparison. The influence of each flavonol and its molar fraction, as well as the interaction between the molar fraction and flavonol typ were tested. Tukey’s post hoc HSD (honestly significant difference) test was performed to test the differences between the groups. Differences were considered statistically significant if *p* ≤ 0.05. The statistical analysis was performed using the software STATISTICA (data analysis software system), version 12.0 (StatSoft, Tulsa, OK, USA).

## 3. Results and Discussion

### 3.1. Antioxidant Activity of Flavonols

Most studies of the antioxidant activity of different flavonols have been performed in oils [50], emulsions [51,52,53], and low density lipoproteins (LDL) [54,55,56]. The antioxidant activity of MCE has been shown to be more effective than QUE in oils in a study of oxidative processes by chelating metal ions [57], while the opposite effect has been found in fish phospholipid liposomes by measuring the amount of thiobarbituric acid-reactive substances (TBARS) produced [58]. We examined the antioxidant activity of three structurally different flavonols, QUE, MCE, and MCI, in a PBS buffer solution (pH = 7.4) using EPR spectroscopy. The EPR signal intensity of the DPPH radicals decreased upon flavonol addition for 20 min in recording intervals of 0.5 min and 1.0 min (Figure 3a). The difference between the integral EPR intensities of the reference and the samples in the 20th min characterized the amount of radicals scavenged by the various components present in the sample acting as radical scavengers.

MCE showed the highest activity in scavenging DPPH radicals, while QUE showed the lowest. This is in agreement with the reported study of the antioxidant activity of flavonols in liposomes [59]. According to Figure 3b, we can order the flavonol samples in relation to their total radical scavenging activity: MCE > MCI > QUE.

### 3.2. Products of Lipid Peroxidation with and without Inserted Flavonols from Mass Spectrometry (MS)

To examine the extent of the lipid peroxidation process initiated by the addition of hydrogen peroxide and iron(II) ions to the liposome suspension, we performed MS and FTIR-ATR spectroscopy measurements. Lipid peroxidation can lead to structural and dynamic changes of the membrane, which can cause an increase in permeability and a change in the lipid ordering and fluidity as well as the bilayer thickness [60,61,62,63,64]. Aldehydes and ketones are products of lipid peroxidation and are known to play a significant role in many human disorders [65].

In our experiments, we used High-Resolution Mass Spectrometry (HR-MS) [66] to qualitatively identify the products after induced lipid peroxidation. The lipid peroxidation of multilamellar DOPC liposomes and derivatization revealed signals indicating short-chain and long-chain LPPs. (Appendix A). Small aliphatic LPPs were detected, as well as several carbonylated lipid species (high molecular weight). It was possible to observe 11 CHH-derivatized carbonyls with different numbers of carbon atoms in the oxidized DOPC liposomes (Appendix A). Among the detected products were butanal, hexanal, and a few long-chain products which have already been reported [67,68]. In addition to the previously mentioned, we also detected acrolein, which is a known neurotoxin produced by lipid peroxidation [69,70] and which has been shown to evoke physiological responses at low concentrations [71].

Based on the data obtained for CHH-derivatized LPPs generated by the oxidation of DOPC, we extended our analytical approach to the derivatized oxidized DOPC liposomes with inserted flavonols present in three different molar fractions (*x* = 0.01, 0.05 and 0.1). Figure 4 shows the derivatized LPPs of DOPC without incorporated flavonoids, with three selected products displayed in smaller inset plots. These products were selected as an example because their peaks gradually disappeared with the increase in the molar fraction of quercetin (as shown in Figure 5). DOPC liposomes with inserted QUE were chosen as an example because the other two flavonols exhibited a stronger inhibitory effect at lower molar fractions (Appendix A). Therefore, the total number of LPPs with different numbers of carbon atoms was determined for each sample and used to evaluate and compare the antioxidative effect of all the flavonols at all molar fractions. In the case of the most hydrophilic flavonol MCI (having a log*P* of 0.5, in contrast to 1.5 and 1.2 for QUE and MCE, respectively [72], the number of observed LPPs decreases the most (to 1, 5, 3 for *x* = 0.01, 0.05 and 0.1, respectively). Furthermore, in the case of the more hydrophobic MCE, the number of LPPs decreases less (to 4, 4, 4 for *x* = 0.01, 0.05 and 0.1, respectively). Finally, for the most hydrophobic QUE, this decrease is the least significant (to 7, 4, 5 for *x* = 0.01, 0.05 and 0.1, respectively). It is known that more hydrophobic antioxidants are distributed in the lipophilic part of the membranes and lipoproteins [64]. In contrast, hydrophilic flavonols are located near the surface of the membrane. Our results confirm the previously reported observations that the rate of radical scavenging within the membrane decreases as the radical goes deeper into the interior of the membrane [73]. That implies that the flavonols located within the membrane (the more hydrophobic ones) scavenge radicals with a lower efficiency than those placed closer to the surface. Given that MCI is located closer to the polar region of the DOPC membrane (closer to the surface), it is more exposed to the radicals entering the liposome from aqueous media. This antioxidative effect of three different flavonols inserted in the liposomes is different than that obtained by measuring the antioxidant activity of flavonols in solution by EPR. Specifically, it appears that when incorporated in the membranes, MCI displays a higher protection than MCE, which can be correlated with their positions inside the lipid bilayer. This indicates that the antioxidative activity of flavonols incorporated in the membrane is different than of those in solution, since the protective effect of antioxidants depends not only on their structure but their location within the bilayer as well. Therefore, in addition to the EPR antioxidative activity assay, the positioning of the flavonols inside membranes should also be taken into account.

Finally, acrolein vanishes in all samples except one (*x* = 0.1 MCE) with incorporated flavonols, which is of great biological significance due to its known neurotoxic properties, as mentioned previously [71].

### 3.3. FTIR-ATR Spectroscopy

#### 3.3.1. FTIR Spectrum of DOPC Lipid Film

To further analyse and try to quantify the extent of the lipid peroxidation process in the presence of flavonols, we performed FTIR-ATR spectroscopy measurements. Figure 6 shows the FTIR-ATR spectrum of DOPC. The assignment of the bands to the specific functional group vibration modes has been made by comparing the obtained spectra with literature data [42,74,75,76].

The 3000–2800 cm^−1^ region in the infrared spectra usually corresponds to ν(C–H) vibrations of the methylene and methyl groups in aliphatic molecules. Two bands with the highest absorbance in that area are near 2924 and 2854 cm^−1^, and they are assigned to methylene antisymmetric and symmetric stretching vibrations, respectively. The band observed near 3006 cm^−1^ corresponds to the stretching vibration of the =C–H cis-olefinic groups, while the shoulder near 2959 cm^−1^ is attributed to the antisymmetric stretching of methyl groups. A strong band near 1737 cm^−1^ is assigned to carbonyl band stretching, while the weaker band around 1652 cm^−1^ is associated with cis C=C bond stretching. Methylene bending bands (scissoring) can be found around 1466 cm^−1^, and near 1377 cm^−1^ is the band that can be attributable to the symmetrical bending vibrations of methyl groups. The vibrational bands in the region below 1300 cm^−1^ arise from the polar headgroup. The antisymmetric PO_2_^−^ stretching is located around 1246 cm^−1^, while the symmetric stretching is located around 1091 cm^−1^. The band around 1172 cm^−1^ corresponds to single-bond C–O stretching. Antisymmetric and symmetric choline stretching (C–N^+^–C) is located near 969 cm^−1^ and 926 cm^−1^, respectively.

#### 3.3.2. The Extent of Lipid Peroxidation

The extent of lipid peroxidation was quantified by calculating the inhibition (*R_i_*) according to the Equations (1)–(3). As shown in Appendix B, *R*_i_ corresponds to the inhibition only if the incorporation of flavonols does not change the absorption coefficient of the present species as well as the ratio of formed LPPs significantly. However, it can be observed that at the highest flavonol ratio this effect cannot be neglected, since the absolute values of *R*_i_ are greater than 1. Indeed, the addition of flavonols can alter the membrane properties [47], which leads to a change in the molar absorption coefficient. All three flavonols exhibit an antioxidative effect upon initiated lipid peroxidation (Appendix A, Figure 7). The statistical analysis showed that there is no significant difference (*p* = 0.22) between the obtained results for different types of flavonols and different molar fractions. Although, visually, the largest difference seems to be in the case of QUE (from *x* = 0.01 to *x* = 0.1), this was not confirmed by the statistical analysis due to the large SD. Therefore, a direct comparison of the three flavonols in the context of their inhibitory activity cannot be made. However, the fact that all flavonols at all molar fractions exhibit antioxidative behaviour is in accordance with the MS measurements (Appendix A). 

### 3.4. Characterization of the Structural Changes of DOPC and Flavonol-Inserted DOPC Liposomes Resulting from Induced Lipid Peroxidation

#### 3.4.1. Dynamic Light Scattering (DLS) and Electrophoretic (ELS) Measurements

The sizes (hydrodynamic diameter, *d*_H_) and zeta potentials of extruded DOPC liposomes were determined using DLS and ELS (Table 1). Since lipid peroxidation causes damage to the lipid bilayers followed by the formation of lipid fragments, our goal was to examine whether the loss of lipid material upon peroxidation causes increased electrostatic interactions between liposomes and, consequently, their aggregation. Thus, these results should be treated as a qualitative indication of the presence of low molecular size LPPs. 

The extruded multilamellar liposomes exhibited negative zeta potentials (*ζ* = −3.4 ± 0.6 mV) (Table 1). A slightly lower absolute value of zeta potential for DOPC liposomes in PBS buffer (pH = 7, *ζ* = −2.29 ± 0.54 mV) was obtained in a recent study by Rudolphi-Skórska et al. [77]. In addition, a somewhat higher value of DOPC liposome zeta potential *ζ* = −4.2 mV) has already been reported [78], confirming the reproducibility of the results. The negative zeta potential of the DOPC liposomes in PBS buffer is a consequence of ion binding at *I*_c_ = 0.15 M. Since the flavonol deprotonation constants are p*K*_a_ = 5.87 and 8.48 [79], p*K*_a_ = 6.33 [80], and p*K*_a_ = 5.23 [81] for QUE, MCE, and MCI, respectively, anion species formed by the deprotonation of QUE, MCE, and MCI are predominant at pH 7.4. The observed zeta potential increase in most samples indicated a significant change in the number of charged groups present on the surface of the liposomes. 

A decrease in the absolute zeta potential value observed in DOPC liposomes with MCE (*x* = 0.01 and 0.05) and MCI (*x* = 0.05) is related to flavonol packing within the bilayer and the strong reorganization of the bilayer (Table 1), which indicates different interactions between flavonols and lipid molecules depending on the flavonol hydrophobicity. However, after 1 h of the exposure to H_2_O_2_ + Fe^2+^ ions, the zeta potential mainly shifts towards negative values, indicating the charging of the surface and suggesting a noticeable loss of lipid material. The majority of the LPPs are polar organic compounds that adsorbed on the surface and increased the zeta potential values. In contrast to these, pure DOPC and DOPC with QUE (*x* = 0.01) exhibited a shift towards positive zeta potential values, indicating a loss of surface charge. The observed surface charging induced by lipid peroxidation resembles the results reported for 1-palmitoyl-2-oleoyl-sn-glycero-3-phosphocholine (POPC) liposomes [82,83]. The present results indicate that the specific behavior of the DOPC membrane might be connected to the inner structure of the bilayer, i.e., to the hydrophobic parts of the lipids.

The liposomes prepared from pure DOPC had an average hydrodynamic diameter, *d*_H_ = (1049 ± 140) nm. After the lipid peroxidation, they became smaller, with a *d*_H_ = (932 ± 105) nm. In flavonol-loaded liposomes, particularly those with inserted MCE (*x* = 0.05) and MCI (*x* = 0.1), an increase in hydrodynamic diameter appeared. This indicates that the bilayer disintegration might have led to the simultaneous liposome aggregation effect [84]. Similar behaviour has already been observed in the study of Mosca et al. [82], where the effect of cholesterol and its esterified derivative, cholesteryl stearate, has been discussed in terms of the influence on the size and ζ-potential of liposomes. 

The specific influence of QUE, MCE, and MCI on the liposome surface charging during lipid peroxidation could not be assigned, but it has been confirmed that the reaction leads to a significant loss of material. The observed behaviour was further examined using SAXS. 

#### 3.4.2. Small Angle X-ray Scattering (SAXS)

To additionally investigate changes in the structure of DOPC liposomes with and without inserted flavonols after induced lipid peroxidation, SAXS has been employed on multilamellar liposomes resuspended in iron(II) chloride tetrahydrate and hydrogen peroxide water solutions. In this study, we used DOPC multilamellar vesicles (MLVs). To the best of our knowledge, this is the first study of the peroxidation of liposome bilayers and multilayers with and without built-in flavonols employing this technique. The intensity function of the multilayer shows sharp Bragg peaks (Figure 8a(bottom),b). After the induced lipid peroxidation (Figure 8a(top)), only single bilayers remained (indicated by the broad peak) with minor traces of multilamellar structures, indicating significant damage to the lamellar structure of the liposomes (Appendix A). 

Since we did not observe major differences in the intensity functions of liposomes containing different flavonols, we calculated the electron density function from the first two Bragg peaks for the DOPC liposomes with inserted MCE (*x* = 0.1). The function increases relatively rapidly from the hydrophobic part towards the hydrophilic part and has a period of 6.1 nm, which is the dimension of one symmetric unit. The electron density is lowest at the position of the methyl groups of the DOPC lipid (*r* = 0) and highest at the position of the phosphatidyl headgroups of the DOPC lipid (*r* = ±1.8 nm). The center of the interstitial water layer is located at *r* = ±3.05 nm.

For the peroxidated systems (bilayers with traces of multilayers), which show only a diffuse broad peak between scattering vector *q*, (0.5 < *q* < 2.5) nm^−1^, the GIFT and DECON software were used to calculate the electron density profiles. From the SAXS curves (Figure 8a), it can be seen that there are no significant differences in the electron densities between the different samples. Therefore, to see the differences in the structures before and after the induced lipid peroxidation, we chose the sample that showed the highest protective effect in the MS measurements, namely DOPC with MCI (*x* = 0.1), which appeared as a pure single-bilayer structure in the SAXS curve after peroxidation. The experimental SAXS data of the DOPC_MCI at *x* = 0.1 after the lipid peroxidation is presented in Appendix A, and the calculated *p*(*r*)-function, assuming lamellar symmetry, is shown in Appendix A. From the pair distance function, *p*(*r*), the electron density function was calculated (Appendix A). The electron density profile showed a large negative slope in the hydrocarbon core of the DOPC bilayers from 0 nm to +1 nm.

The effect of lipid peroxidation on the structure of the lipid bilayer was not only observed by comparing the shape of the scattering functions before and after the induced lipid peroxidation (Figure 9a) but also by comparing the electron density functions (Figure 9b). The electron density function for DOPC with MCI (*x* = 0.1) before the induced lipid peroxidation was higher than the one before lipid peroxidation at distances from 0 nm to 0.5 nm from the terminal hydrophobic group. In additon, the maxima corresponding to the phosphatidyl headgroups were shifted towards the bilayer center. This is in an agreement with the study of the 1-palmitoyl-2-linoleoyl-sn-glycero-3-phosphatidylcholine (PLPC) bilayer, in which the total density at the center of the bilayer was increased upon the initiated oxidation, and the maxima were shifted toward the center [85]. Furthermore, when the PLPC liposomes were oxidized, the density at the centre of the bilayer increased, corresponding to the partial interdigitation of the phospholipid acyl-chain terminal methyl segment. In our case, the density at the centre of the bilayer decreased, indicating that the MCI suppressed the aforementioned interdigitation. This observation is another confirmation of flavonol antioxidative and protective acitivity upon initiated peroxidation. Observed changes in the lipid bilayer structure with inserted flavonols following lipid peroxidation provide evidence for minor changes in the membrane hydrocarbon core width. 

#### 3.4.3. Nanomechanical Properties of Supported Lipid Bilayers (SLBs) before and after Induced Lipid Peroxidation

The main task of this research is to investigate the impact of lipid oxidation on the structural properties of SLBs in the absence and presence of flavonols. It has been shown that AFM is an efficient tool to study the formation mechanism of SLBs [47,86] because it provides information about the structure of an adsorbed material and its elastic properties. Flavonol insertion in the membrane induces a deformation of the membrane surface, resulting in changes in the membrane roughness and thickness. On the other hand, the lipid peroxidation induces the fragmentation of membrane lipids as well as structural damage, which should be supressed in the presence of flavonols. In that case, the extent of lipid peroxidation manifests through the alteration of crucial structural parameters. 

The protective role of the flavonols depends on the mutual interactions between lipids and flavonols. The question that has arisen was whether flavonoids, concerning their hydrophobicity and structure, are able to insert laterally and homogeneously in the bilayer. The top views of the morphology of DOPC SLBs with and without inserted flavonols are presented in Figure 10 and Figure 11, and Appendix A. Since the imaging conditions were the same for all the investigated samples, all the observed changes in the morphology of the formed SLBs correspond only to the differences between the samples. The cross-section profiles (Appendix A) and bilayer thicknesses obtained from the jump in the force curves correspond to the single SLB. The homogeneous rough SLB patches are seen irrespective of inserted flavonols.

At the position where the flavonol molecule is located, a small cavity of water molecules is formed, followed by a change in the membrane thickness. The roughness values for each sample were corrected for the convolution effect of the tip [87]. The flavonol insertion and the lipid peroxidation process alter the initial roughness values, as indicated in Figure 12. 

Our fine-structure analysis clearly indicated that the presence of flavonols caused a remarkable increase in the surface roughness. For example, for the control system, i.e., pure DOPC, the roughness is *R*_a_ = (0.060 ± 0.008) nm, while in the presence of glycone MCI, the surface roughness dramatically increased. A further increase in the MCI molar fraction only slightly modified the surface roughness. On the other hand, by insertion of more hydrophobic QUE and MCE, the surface roughness increased, but the effect is lower for QUE, which is the most hydrophobic flavonol (Table 2, Figure 12A). It can be concluded that more hydrophobic flavonols cause smaller changes in the roughness than the hydrophilic ones. Therefore, the surface roughness can be a good indicator of the location of the flavonols within the bilayer. 

Lipid peroxidation causes the fragmentation of lipids and deteorioration of the membrane, which is the effect that should be reflected in the roughness values. Our results showed that after the induced lipid peroxidation, the roughness of DOPC without flavonols dramatically increased. The samples of DOPC SLBs with incorporated flavonols showed a smaller increase or even a decrease in the surface roughness, indicating their role in the preservation of the overall integrity of the membrane. 

The interaction of lipid molecules and flavonols, as well as the lipid peroxidation effects, were further monitored by measuring the elasticity, i.e., Young’s modulus (*E*). The results are summarized in Table 2, Figure 13, and elasticity maps (Appendix A). The elasticity value obtained for the pure DOPC SLB (*E* = 15.4 ± 5.4 MPa) could be attributed to the fluid phase of DOPC and is in an agreement with already reported values [88]. The insertion of QUE and MCE caused only a minor change in the elasticity. In contrast, the insertion of hydrophilic MCI induced a drastic increase in the elasticity, which is in accordance with its presumed location. Due to increased interaction between the phosphate headgroups and MCI, the MCI modifies the orientation of the bilayer dipoles, leading to the increased number of hydrogen bonds. However, AFM images (Figure 10 and Figure 11 and Appendix A) showed that the change in the membrane stiffness was not sufficient to disorganize or destabilize the whole SLB structure by pore formation. These results are in accordance with already reported studies of flavonols and SLBs with aglycone hesperetin and glycone hesperidin [88,89], suggesting different flavonol permeation with respect to their own different hydrophobicity or hydrophilicity. Thus, the bilayer disordering caused by the flavonol presence, as well as their insertion, is in an agreement with their corresponding partition coefficients. 

The distinct increase in the Young’s modulus was observed after the exposure of DOPC SLB to hydrogen peroxide and iron(II) ions. In general, the jump in the elasticity after the induced lipid peroxidation decreases with an increase in the molar fraction of flavonols (Figure 13b, Table 2), indicating the preservation of the membrane elasticity. The only exception is DOPC with MCI at the highest molar fraction, where a decrease in the elasticity is observed after induced lipid peroxidation. Since the value of *E* after peroxidation drops to a similar value as in the case of the pure DOPC liposomes before lipid peroxidation, this could indicate the MCI leaving from the membrane. Another possible explanation for this decrease could be the change in the interactions of the MCI and polar headgroups after the lipid peroxidation. 

During lipid peroxidation, the change in the bilayer thickness appears as a consequence of both strong reorganization inside the lipid bilayer (intedigitation) and the adsorption of surface-active LPP fragments onto the surface of the membrane. The former effect would cause a decrease in the thickness, while the latter would cause an increase. An increase in the membrane thickness of pure DOPC SLBs agrees with the aforementioned adsorption of LPP fragments on the surface. Because of the competing effects, the changes in thickness after induced lipid peroxidation in the liposomes with flavonols does not show a particular trend (Figure 14, Table 2).

### 3.5. Fluidity Change upon Initiated Lipid Peroxidation by EPR

Analysis of the EPR spectra of 5-DSA radicals allowed us to characterize the properties of the spin probe environment. The presence of two or more spectral components with different parameters may give additional information about the processes occurring in the model lipid membranes. The EPR spectra of the 5-DSA spin probe in flavonol samples were recorded in order to establish a correlation between the different concentrations, lipid peroxidation, and EPR parameters of the spin probe. The parameters that can be followed are the ratio between the fast (mobile) (*w*(F)) and slow (immobile) (*w*(*S*)) components and the distance between the outer peaks (2*A_zz_*). The hyperfine coupling constant of a nitroxide (*a_oN_*) probe is sensitive to polarity in the sense that it increases in a polar environment. The lipid peroxidation would alter the bilayer fluidity, as can be seen throughout the modification of the typical order parameter, *S*. Fatty acid residues within the bilayer (detected by MS spectroscopy) also have an average common orientation along the bilayer. Owing to the axial anisotropy of the nitroxide magnetic parameter *A*, two different EPR spectra are observed according to the orientation of the SLB normal with respect to the direction of the external magnetic field (assuming the common orientation of the 5-DSA). Therefore, the observation of two distinct EPR spectra upon changing the sample orientation reveals internal bilayer spatial ordering. In addition, it is expected that the presence of oxygenated groups, aldehyde or carboxylic group, disturbs the fatty acid ordering. However, in this case, the consequent disordering of the spin label becomes visible as the EPR spectral anisotropy changes. All the EPR spectra were simulated (Figure 15), and the spectral parameters are shown in Table 3.

The S-value of the pure DOPC bilayer is 1.0, while *S* falls to 0.33 after the addition of hydrogen peroxide and iron(II) ions to the liposome suspension, confirming a strong increase in the fluidity (Appendix A). A similar increase in the fluidity of the DOPC lipid bilayer has been reported by Tai and coworkers [60]. The insertion of flavonols into the DOPC bilayer also caused a decrease in the rigidity of the bilayer, but the effect is minor with all the used flavonols. On the other hand, the variation in 2*A_ZZ_*, chosen as the most fluidity-sensitive parameter of EPR spectra with the molar fraction of flavonols reveals the highest decrease in fluiditiy at all the tested QUE molar fractions in comparison to two other flavonols. The response of the system to lipid peroxidation change is shown in the shift to higher 2*A_ZZ_* values, turning the bilayer more rigid in the whole molar fraction range of flavonols. Only one exception was observed in the case of QUE at the molar fraction x = 0.1. This behavior is consistent with our antioxidant activity analysis of the three flavonols revealing the lower activity of QUE, and are in good agreement with the study of the membrane “fluidity”, which has been repressed in microsomes treated with α-tocopherol before but not after peroxidation, suggesting that the inhibitory effect was due to its antioxidant activity [89]. However, our observations are consistent with the possibility of a number of hydrogen bonds occurring between the hydroperoxyl groups.

The well-documented modulation of the lipid bilayer fluidity by lipid peroxidation can lead to the conclusion that lipid peroxidation affects membrane fluidity through the increase in the number of hydroxy and oxo-C-chains, on one side, and the chain shortening on the other. Besides this, the non-negligible degree of inhibition of the observed modulation by flavonols within the lipid bilayer has been confirmed in this study.

## 4. Conclusions

We have used a multi-technique approach to study the structural changes of the model lipid membrane caused by lipid peroxidation and the antioxidative role of three structurally and chemically different flavonols, QUE, MCE, and MCI. 

Lipid peroxidation leads to structural alterations that violate the membrane integrity, as indicated by the changes in the crucial properties of the membranes, such as elasticity, surface roughness, thickness, and fluidity. All these properties are essential for the function of biological membranes.

The antioxidative activity of the three investigated flavonols strongly depends on the environment in which they are located. Specifically, when the antioxidative assay was performed in the solution, MCE showed the highest activity in the scavenging of free radicals, while the most hydrophobic flavonol, QUE, showed the lowest. In contrast to this, the comparison between the flavonols when they were incorporated within the liposome membrane displayed different results. The most hydrophilic flavonol, MCI, showed a higher protection than QUE and MCE, which are more hydrophobic and located deep within the bilayer. This was concluded based on the fact that, while the number of identified peroxidation products (taken as the measure of flavonol inhibition) decreased with an increase in the molar fraction for all the flavonols, the effect was the most pronounced in the case of MCI. Since MCI is located closer to the surface of the membrane, oriented towards the aqueous medium, it is more exposed to the incoming radicals and is able to scavenge them before they reach the reactive site. 

The loss of multilamellar structure and the loss of lipid material upon lipid peroxidation indicated multiple surface processes and the rearrangement of the membrane. As a result of these processes, changes in the surface roughness and elasticity were observed. Since these changes are less pronounced in the presence of flavonols, it can be concluded that they preserved the supramolecular and mechanical properties of the membrane. Finally, the significant degree of inhibition of peroxidation process of all investigated flavonols has also been confirmed by the preservation of the bilayer fluidity. 

Our study suggests that all the techniques employed can be used as a highly valuable tool in other biomedical applications aimed at screening and monitoring the lipid peroxidation effects at the cellular level. Furthermore, it was demonstrated that, when studying the protective activity of various antioxidants, it is necessary to consider their environment along with their chemical properties. 

## Figures and Tables

**Figure 1 antioxidants-09-00430-f001:**
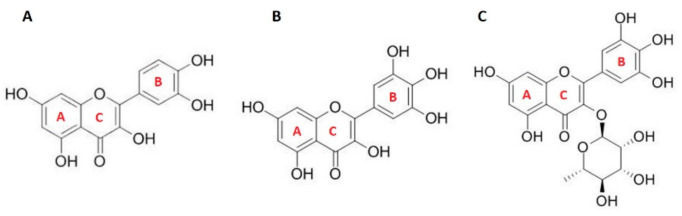
Structure of flavonols: (**A**) quercetin (QUE); (**B**) myricetin (MCE); (**C**) myricitrin (MCI).

**Figure 2 antioxidants-09-00430-f002:**
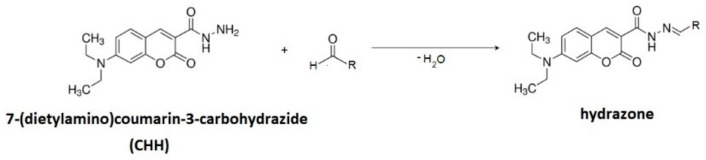
Scheme of the derivatization with 7-(diethylamino)coumarin-3-carbohydrazide into hydrazone.

**Figure 3 antioxidants-09-00430-f003:**
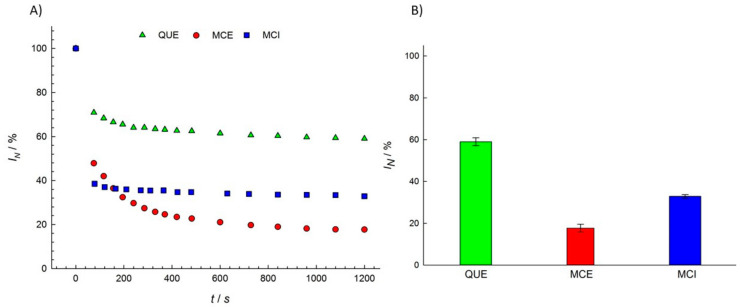
(**A**) The normalized EPR signal intensity of the DPPH radical measured as a function of the reaction time t; (**B**) the normalized EPR signal intensity (*I*_N_) of the DPPH radical at a *t* = 1200 s reaction time for various flavonol solutions (1 vol %). The statistical analysis showed that all the groups (QUE, MCE, MCI) significantly differ from each other.

**Figure 4 antioxidants-09-00430-f004:**
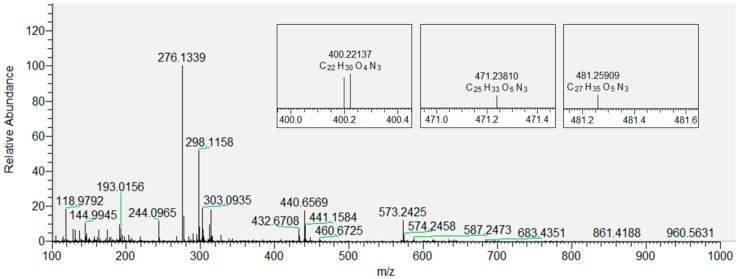
Mass spectrum of LPP ions of DOPC with zoomed-in regions corresponding to the molecular ions of three derivatized products: hydroxy-octenal (left), hydroxy-oxo-undecenoic acid (middle), and oxo-tridecadienoic acid (right).

**Figure 5 antioxidants-09-00430-f005:**
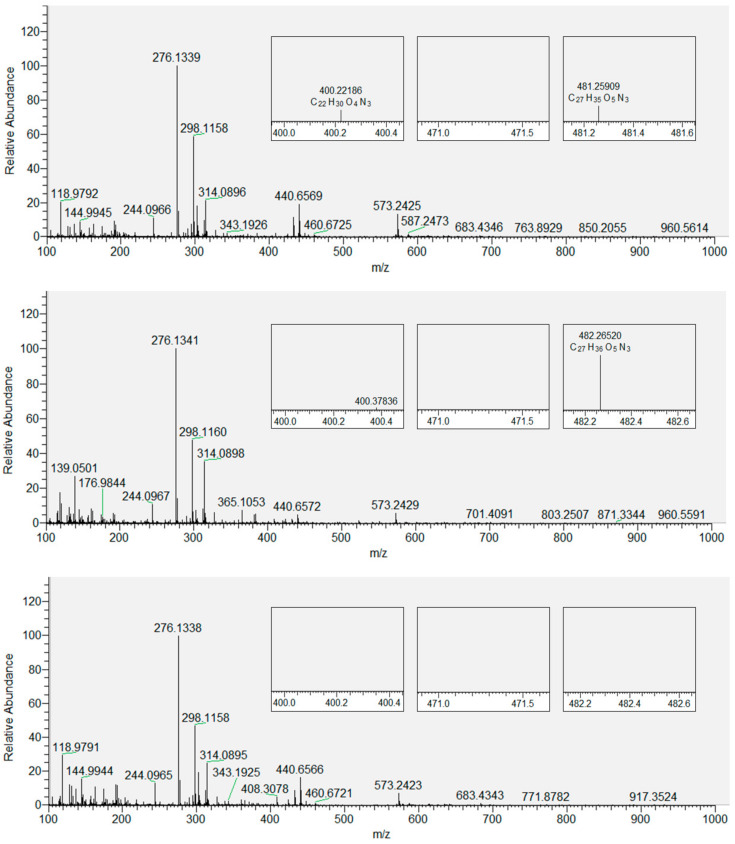
Mass spectra of derivatized LPP ions of DOPC with QUE: top: *x* = 0.01; middle: *x* = 0.05; bottom: *x* = 0.1. Zoomed-in regions correspond to the molecular ions of three derivatized products: hydroxy-octenal (left), hydroxy-oxo-undecenoic acid (middle), and oxo-tridecadienoic acid (right).

**Figure 6 antioxidants-09-00430-f006:**
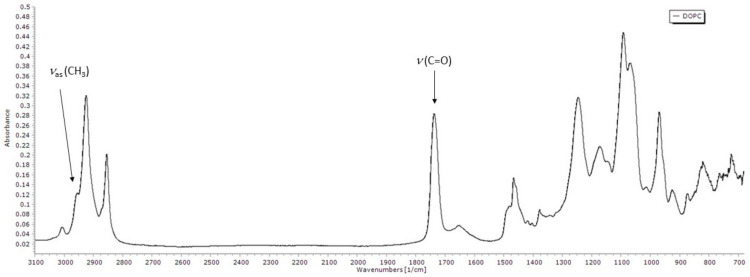
FTIR-ATR spectrum of DOPC with two analyzed bands pointed out.

**Figure 7 antioxidants-09-00430-f007:**
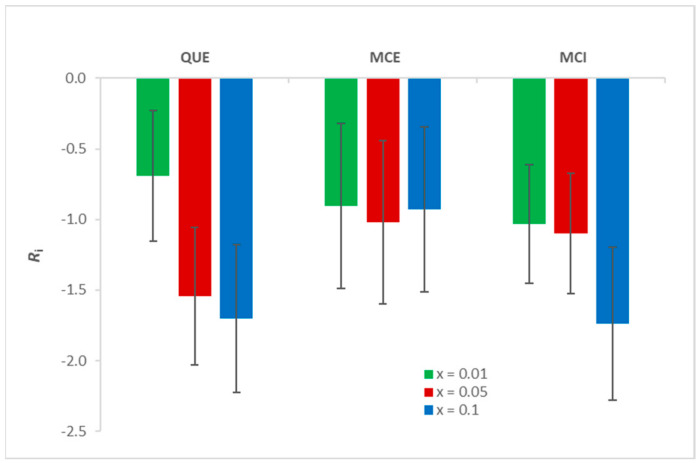
The inhibition of lipid peroxidation (*R_i_*) for three molar fractions of QUE, MCE, and MCI.

**Figure 8 antioxidants-09-00430-f008:**
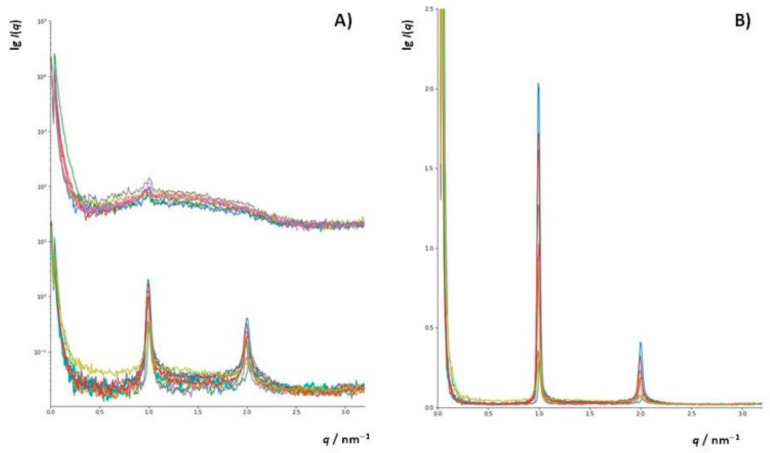
(**A**) SAXS curves (lg(*I*) vs. *q*) (raw data, not background-subtracted) of lipid solutions, separated (shifted verically) according to the nanostructure: (bottom) multilamellar structure (top), bilayers, and the rest of the material. (**B**) SAXS curves in linear plots (*I* vs. *q*) of lipid solutions showing the multilamellar nanostructure.

**Figure 9 antioxidants-09-00430-f009:**
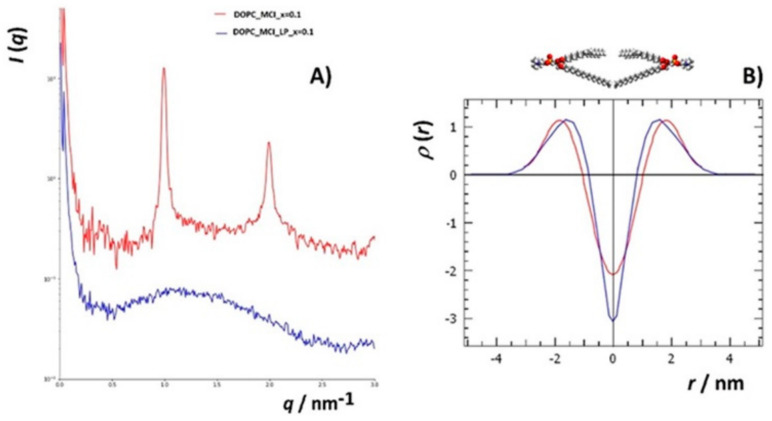
Comparison of the electron densities (**B**, normalized to the peak maximum) of DOPC_MCI at molar fraction *x* = 0.1 (red) and DOPC_MCI_LP at *x* = 0.1 (blue) obtained from their respective SAXS curves (**A**, background subtracted).

**Figure 10 antioxidants-09-00430-f010:**
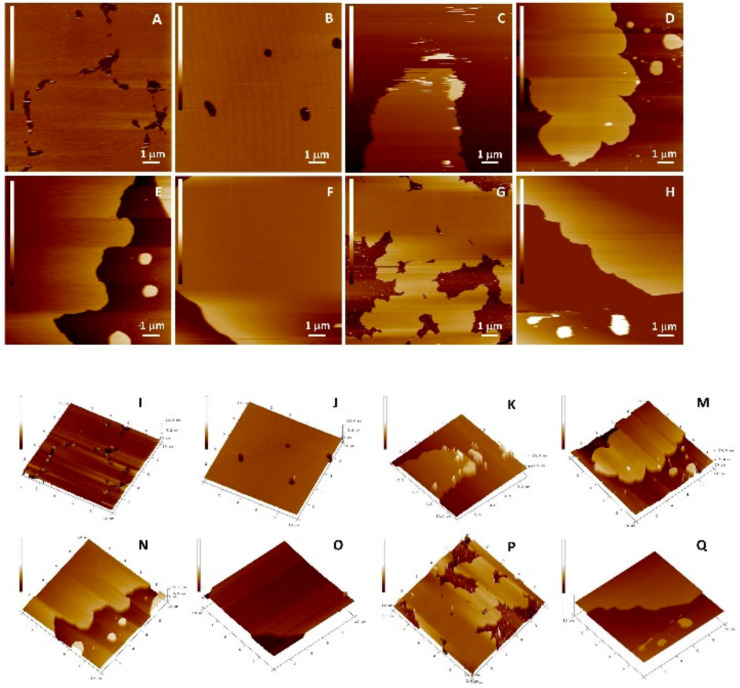
Top view of height AFM images on the model (DOPC) SLB without (*x* = 0) and with inserted QUE at different molar fractions before the induced lipid peroxidation. 2D view: (**A**) DOPC (*x* = 0); (**B**) (*x* = 0.01); (**C**) (*x* = 0.05); and (**D**) (*x* = 0.10). 3D view: (**I**) DOPC (*x* = 0); (**J**) (*x* = 0.01); (**K**) (*x* = 0.05); and (**M**) (*x* = 0.10). After the induced lipid peroxidation, 2D view: (**E**) DOPC (*x* = 0); (**F**) (*x* = 0.01); (**G**) (*x* = 0.05); and (**H**) (*x* = 0.10). 3D view: (**N**) DOPC (*x* = 0); (**O**) (*x* = 0.01); (**P**) (*x* = 0.05); and (**Q**) (*x* = 0.10). Scales of all images are 6 nm.

**Figure 11 antioxidants-09-00430-f011:**
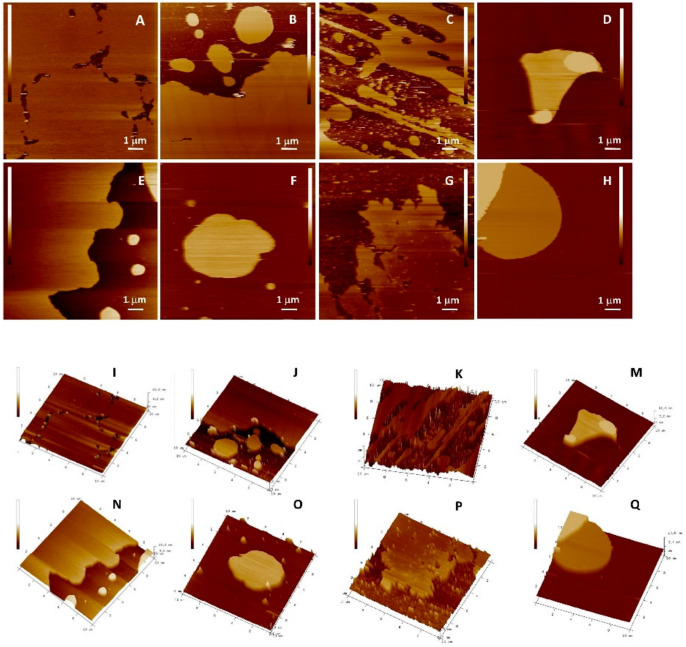
Top view of height AFM images on the model (DOPC) SLB without (*x* = 0) and with inserted MCE at different molar fractions before the induced lipid peroxidation. 2D view: (**A**) DOPC (*x* = 0); (**B**) (*x* = 0.01); (**C**) (*x* = 0.05); and (**D**) (*x* = 0.10). 3D view: (**I**) DOPC (*x* = 0); (**J**) (*x* = 0.01); (**K**) (*x* = 0.05); and (**M**) (*x* = 0.10). after the induced lipid peroxidation, 2D view: (**E**) DOPC (*x* = 0); (**F**) (*x* = 0.01); (**G**) (*x* = 0.05); and (**H**) (*x* = 0.10). 3D view (**N**) DOPC (*x* = 0); (**O**) (*x* = 0.01); (**P**) (*x* = 0.05); and (**Q**) (*x* = 0.10). Scales of all images are 6 nm.

**Figure 12 antioxidants-09-00430-f012:**
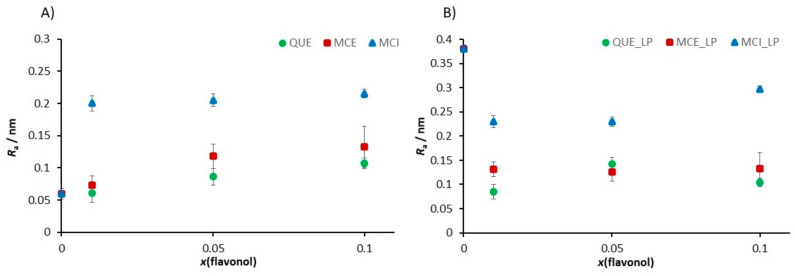
The change in the roughness value induced (**A**) by the insertion of flavonols in dependence of the molar fraction of inserted flavonols and (**B**) by the induced lipid peroxidation process in the SLB as a function of the molar fraction of inserted flavonols.

**Figure 13 antioxidants-09-00430-f013:**
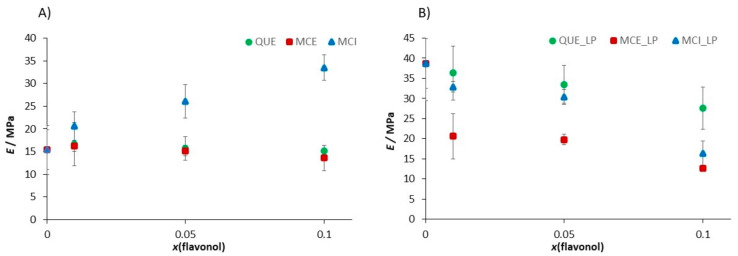
The change of the Young’s modulus value induced (**A**) by the insertion of the flavonols in dependence of the molar fraction of inserted flavonols and (**B**) by the induced lipid peroxidation process in the SLB as a function of the molar fraction of inserted flavonols.

**Figure 14 antioxidants-09-00430-f014:**
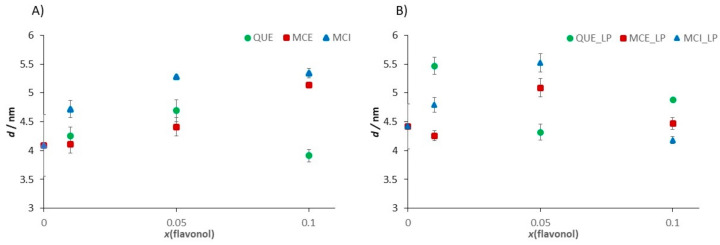
The change in the bilayer thickness value induced (**A**) by the insertion of the flavonols in dependence of the molar fraction of inserted flavonols and (**B**) by the induced lipid peroxidation process in the SLB as a function of the molar fraction of inserted flavonols.

**Figure 15 antioxidants-09-00430-f015:**
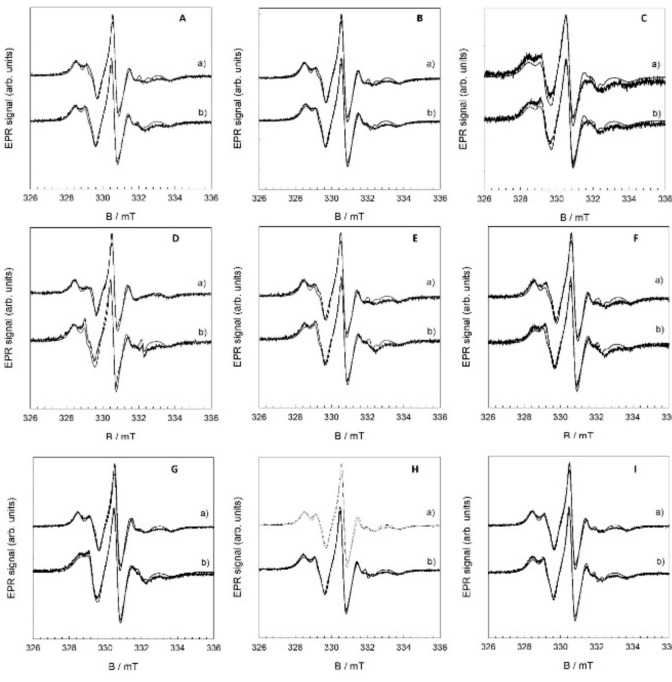
EPR spectra before (a) and after the induced lipid peroxidation (b) of DOPC liposomes with inserted flavonols QUE (**A**, *x* = 0.01; **B**, *x* = 0.05; **C**, *x* = 0.1), MCE (**D**, *x* = 0.01; **E**, *x* = 0.05; **F**, *x* = 0.1), and MCI (**G**, *x* = 0.01; **H**, *x* = 0.05; **I**, *x* = 0.1).

**Table 1 antioxidants-09-00430-t001:** Changes in the average hydrodynamic diameter (*d*_H_) and zeta potential (*ζ*) of liposomes during the insertion of flavonols and induced lipid peroxidation.

Sample	*d*_H_/nm	*ζ*/mV
DOPC	1049 ± 140	−3.4 ± 0.6
DOPC/H_2_O_2_ + Fe^2+^	932 ± 105	−2.8 ± 0.9
DOPC/QUE (*x* = 0.01)	148 ± 48	−3.9 ± 0.3
DOPC/QUE (*x* = 0.01)/H_2_O_2_ + Fe^2+^	1319 ± 57	−2.9 ± 0.6
DOPC / QUE (*x* = 0.05)	1258 ± 117	−3.8 ± 0.4
DOPC/QUE (*x* = 0.05)/H_2_O_2_ + Fe^2+^	1351 ± 205	−4.9 ± 0.3
DOPC/QUE (*x* = 0.1)	202 ± 79	−5.7 ± 0.5
DOPC/QUE (*x* = 0.1)/H_2_O_2_ + Fe^2+^	487 ± 64	−7.1 ± 0.3
DOPC/MCE (*x* = 0.01)	107 ± 16	−2.4 ± 0.2
DOPC/MCE (*x* = 0.01)/ H_2_O_2_ + Fe^2+^	577 ± 159	−3.6 ± 0.4
DOPC/MCE (*x* = 0.05)	72 ± 6	−2.5 ± 0.5
DOPC/MCE (*x* = 0.05)/H_2_O_2_ + Fe^2+^	1031 ± 152	−3.6 ± 0.2
DOPC/MCE (*x* = 0.1)	203 ± 119	−3.7 ± 0.5
DOPC/MCE (*x* = 0.1)/H_2_O_2_ + Fe^2+^	406 ± 105	−9.8 ± 2.2
DOPC/MCI (*x* = 0.01)	163 ± 87	−3.5 ± 0.3
DOPC/MCI (*x* = 0.01)/H_2_O_2_ + Fe^2+^	180 ± 42	−4.0 ± 0.6
DOPC/MCI (*x* = 0.05)	222 ± 42	−2.8 ± 0.2
DOPC/MCI (*x* = 0.05)/H_2_O_2_ + Fe^2+^	266 ± 44	−4.3 ± 0.2
DOPC/MCI (*x* = 0.1)	155 ± 30	−3.9 ± 0.5
DOPC/MCI (*x* = 0.1)/H_2_O_2_ + Fe^2+^	1746 ± 654	−8.7 ± 1.2

**Table 2 antioxidants-09-00430-t002:** The effect of the insertion of flavonols and induced lipid peroxidation on the roughness (*R*_a_), Young’s Modulus (*E*), and bilayer thickness (*d*) of the model DOPC SLB (n = 6).

Sample	*R*_a_/nm	*E/*MPa	*d*/nm
DOPC	0.060 ± 0.008	15.4 ± 5.4	4.09 ± 0.53
DOPC/H_2_O_2_ + Fe^2+^	0.381 ± 0.018	38.7 ± 6.2	4.42 ± 0.39
DOPC/QUE (*x* = 0.01)	0.061 ± 0.015	16.6 ± 4.8	4.25 ± 0.16
DOPC/QUE (*x* = 0.01)/H_2_O_2_ + Fe^2+^	0.085 ± 0.026	36.3 ± 6.7	5.47 ± 015
DOPC/QUE (*x* = 0.05)	0.086 ± 0.013	15.7 ± 2.6	4.69 ± 0.19
DOPC/QUE (*x* = 0.05)/H_2_O_2_ + Fe^2+^	0.143 ± 0.018	33.5 ± 4.7	4.32 ± 0.14
DOPC/QUE (*x* = 0.1)	0.107 ± 0.008	15.2 ± 1.1	3.91 ± 0.11
DOPC/QUE (*x* = 0.1)/H_2_O_2_ + Fe^2+^	0.104 ± 0.024	27.6 ± 5.3	4.88 ± 0.03
DOPC/MCE (*x* = 0.01)	0.073 ± 0.015	16.2 ± 1.2	4.11 ± 0.15
DOPC/MCE (*x* = 0.01)/H_2_O_2_ + Fe^2+^	0.131 ± 0.034	20.6 ± 5.6	4.26 ± 0.09
DOPC/MCE (*x* = 0.05)	0.118 ± 0.019	15.2 ± 1.1	4.41 ± 0.16
DOPC/MCE (*x* = 0.05)/H_2_O_2_ + Fe^2+^	0.126 ± 0.019	19.8 ± 1.3	5.09 ± 0.16
DOPC/MCE (*x* = 0.1)	0.133 ± 0.032	13.6 ± 2.8	5.14 ± 0.02
DOPC/MCE (*x* = 0.1)/H_2_O_2_ + Fe^2+^	0.133 ± 0.030	12.6 ± 0.1	4.78 ± 0.10
DOPC/MCI (*x* = 0.01)	0.200 ± 0.012	20.6 ± 3.2	4.72 ± 0.15
DOPC/MCI (*x* = 0.01)/H_2_O_2_ + Fe^2+^	0.230 ± 0.023	32.9 ± 1.3	4.79 ± 0.13
DOPC/MCI (*x* = 0.05)	0.205 ± 0.094	26.1 ± 3.7	5.28 ± 0.02
DOPC/MCI (*x* = 0.05)/H_2_O_2_ + Fe^2+^	0.230 ± 0.003	30.4 ± 1.8	5.52 ± 0.16
DOPC/MCI (*x* = 0.1)	0.215 ± 0.006	33.5 ± 2.8	5.34 ± 0.08
DOPC/MCI (*x* = 0.1)/H_2_O_2_ + Fe^2+^	0.297 ± 0.005	16.3 ± 3.1	4.18 ± 0.06

**Table 3 antioxidants-09-00430-t003:** Spectral parameters for DOPC liposomes with and without inserted flavonols before and after starting the lipid peroxidation process (*T* = 291 K).

Sample	*2A_ZZ_*/G	*a_oN_*/G	*w(*S)/%	*w(*F)/%	*S*
DOPC	58.32	-	100	-	1
DOPC + H_2_O_2_ + Fe^2+^	58.20	15.99	32.94	67.06	0.33
DOPC + QUE (*x* = 0.01)	58.97	14.89	91.34	8.64	0.91
DOPC + QUE (*x* = 0.01) + H_2_O_2_ + Fe^2+^	60.39	14.62	94.30	5.70	0.94
DOPC + QUE (*x* = 0.05)	59.68	15.00	89.43	10.57	0.89
DOPC + QUE (*x* = 0.05) + H_2_O_2_ + Fe^2+^	60.01	14.29	96.23	3.77	0.96
DOPC + QUE (*x* = 0.1)	66.40	13.98	93.78	6.22	0.94
DOPC + QUE (*x* = 0.1) + H_2_O_2_ + Fe^2+^	64.72	14.15	92.39	7.61	0.92
DOPC + MCE (*x* = 0.01)	58.37	-	100	-	1
DOPC + MCE (*x* = 0.01) + H_2_O_2_ + Fe^2+^	60.91	15.79	75.01	24.99	0.75
DOPC + MCE (*x* = 0.05)	60.18	15.08	95.52	4.48	0.96
DOPC + MCE (*x* = 0.05) + H_2_O_2_ + Fe^2+^	61.45	14.92	92.80	7.20	0.93
DOPC + MCE (*x* = 0.1)	60.65	14.97	92.64	7.36	0.93
DOPC + MCE (*x* = 0.1) + H_2_O_2_ + Fe^2+^	62.06	15.05	91.50	8.50	0.92
DOPC + MCI (*x* = 0.01)	59.56	13.87	94.28	5.72	0.94
DOPC + MCI (*x* = 0.01) + H_2_O_2_ + Fe^2+^	62.09	13.89	93.81	6.19	0.94
DOPC + MCI (*x* = 0.05)	60.03	14.02	93.52	6.48	0.94
DOPC + MCI (*x* = 0.05) + H_2_O_2_ + Fe^2+^	61.57	13.91	95.32	4.68	0.95
DOPC + MCI (*x* = 0.1)	60.55	14.22	94.16	5.84	0.94
DOPC + MCI (*x* = 0.1) + H_2_O_2_ + Fe^2+^	62.31	14.09	94.40	5.60	0.95

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
