# Peer review of "The Structural Integrity of the Model Lipid Membrane during Induced Lipid Peroxidation: The Role of Flavonols in the Inhibition of Lipid Peroxidation"

_antioxidants, 2020, doi:10.3390/antiox9050430_

Round 1
Reviewer 1 Report
The manuscript investigated, using an in vitro model, the structure/characteristics of lipid components of membranes after induction of lipid peroxidation, and the ability of flavonols in preventing lipid peroxidation and consequent alterations.
The research investigated several aspects of the phenomenon using different investigation techniques. It evidenced that the model lipid membrane could be useful in investigating the antioxidant properties of possible antioxidant substances. Anyway, some criticisms are present.
- The liposome prepared and used during the experiments contains 1,2-dioleoyl-sn-glycero-17 3-phosphocholine (DOPC), therefore the fatty acid undergoing lipid peroxidation is a monounsaturated fatty acid. It is well known that the main substrates of lipid peroxidation process are PUFA, as also reported by authors in Introduction (lines 42-43). For this reason, the study needs to be completed by analysing the effect of flavonols on liposomes containing PUFAs belonging to both n-3 or n-6 families, since they show a different susceptibility to lipid peroxidation.
The importance of including PUFA-containing liposomes in the study is also supported by the observation that among the products of lipid peroxidation of DOPC liposomes (Table 1S) the more reactive lipidic aldehydes, HHE and HNE, are not present.
- Lipid peroxidation has been induced by the addition of hydrogen peroxide and iron(II) ions. Other more physiological conditions inducing lipid peroxidation have to be used, for example the Fe2+/ascorbate system.
- The relevance of the reported results could be improved by comparing the effects of flavonols with those of other antioxidants in the same experimental conditions.
Minor criticisms
Introduction, line 39. The statement “Lipid peroxidation occurrence in the human body is a cause for oxidative stress” has to be modified because lipid peroxidation is a consequence of oxidative stress.
Introduction, line 46-47. In the statement “…the response of the target cell” has to be clarified.
Since a long section “Results and Discussion” is present, the Conclusion has to be shortened and focused on the significant findings of the research, this avoiding repetitions.
Author Response
Dear Reviewer,
thank you for your comments and suggestions. We have revised our manuscript and implemented them. In the attachment, please find the revised manuscript where all the changes are already implemented (for easier reading). The specified line numbers in the answers to your comments correspond to the version with implemented changes.
Thank you again for consideration of our revised manuscript.
Best regards,
Suzana Šegota, PhD, Division of Physical Chemistry
Ruđer Bošković Institute
Bijenička cesta 54,
HR-10 000 Zagreb, Croatia
E-mail: ssegota@irb.hr
Reviewer #1: The manuscript investigated, using an in vitro model, the structure/characteristics of lipid components of membranes after induction of lipid peroxidation, and the ability of flavonols in preventing lipid peroxidation and consequent alterations.
The research investigated several aspects of the phenomenon using different investigation techniques. It evidenced that the model lipid membrane could be useful in investigating the antioxidant properties of possible antioxidant substances. Anyway, some criticisms are present.
Question: The liposome prepared and used during the experiments contains 1,2-dioleoyl-sn-glycero-17 3-phosphocholine (DOPC), therefore the fatty acid undergoing lipid peroxidation is a monounsaturated fatty acid. It is well known that the main substrates of lipid peroxidation process are PUFA, as also reported by authors in Introduction (lines 42-43). For this reason, the study needs to be completed by analysing the effect of flavonols on liposomes containing PUFAs belonging to both n-3 or n-6 families, since they show a different susceptibility to lipid peroxidation. The importance of including PUFA-containing liposomes in the study is also supported by the observation that among the products of lipid peroxidation of DOPC liposomes (Table 1S) the more reactive lipidic aldehydes, HHE and HNE, are not present.
Answer: Like in other unsaturated acids, the oxidation of a double bond in monounsaturated fatty acid occurs by radicals already present in the system, or is initiated by radical source brought into the system. Carbon-centered radicals then rapidly react with oxygen in the air to start free radical chain reaction. However, its initiation phase with DOPC is slower due to absence of methylene bridge between two double bonds. Hydrogen abstraction from such a bridge by a radical is much easier than in methylene groups of monosaturated fatty acids. Thus, the mechanism of lipid peroxidation of DOPC remains essentially the same as for lipids made of PUFA, but the products will likely be different in type and quantity, which can explain the absence of HNE and HHE. It is thus immediately clear that one of the main goals in this work is in establishing the limits of the employed analytical methods. This has also been clarified in the introduction (lines 121-126), with the relevant reference added. Experiments concerning PUFA and the comparison with monounsaturated fatty acids will be done in the future research.
Question: Lipid peroxidation has been induced by the addition of hydrogen peroxide and iron(II) ions. Other more physiological conditions inducing lipid peroxidation have to be used, for example the Fe2+/ascorbate system. The relevance of the reported results could be improved by comparing the effects of flavonols with those of other antioxidants in the same experimental conditions.
Answer: The explanation for the choice of our initiation system has been added to the section 2.2., along with relevant references (lines 170-183). Reported results, discussion and the conclusion were modified to further clarify the relevance of the results.
Minor criticisms
Question: Introduction, line 39. The statement “Lipid peroxidation occurrence in the human body is a cause for oxidative stress” has to be modified because lipid peroxidation is a consequence of oxidative stress.
Answer: As suggested, we made corrections in the manuscript.
Question: Introduction, line 46-47. In the statement “…the response of the target cell” has to be clarified.
Answer: As suggested, we made corrections in the manuscript in order to clarify (lines 48-49), with relevant references added.
Question: Since a long section “Results and Discussion” is present, the Conclusion has to be shortened and focused on the significant findings of the research, this avoiding repetitions.
Answer: As suggested, conclusion was drastically shortened and rephrased, and significant findings were pointed out.

Reviewer 2 Report
The structural integrity of the model lipid membrane 2 during induced lipid peroxidation: the role of 3 flavonols in the inhibition of lipid peroxidation
Corresponding author: Suzana Šegota
Authors: Anja Sadžak, Janez Mravljak, Nadica Maltar-Strmečki, Zoran Arsov, Goran Baranović, Ina Erceg , Manfred Kriechbaum, Vida Strasser, Jan Přibyl
General Summary: This study has a very interesting premise. The authors set out to determine the physical and biochemical characteristics of lipid membranes that are potentially protected from oxidative damage by quercetin, myricetin, and myricitrin, flavonols with established antioxidant and other bioactivity. They use very sophisticated chemistry instrumentation and methods to answer their questions. However, after a relatively strong Introduction section, the manuscript is almost impossible to decipher. The Results/Discussion section is very explanation-heavy but does not connect important concepts back to the study’s results. The authors need to focus on and highlight their data more, rather than extensive descriptions that do not relate to flavonols. Also, the data needs to be made much more understandable. The Methods section is very descriptive, but these details don’t need to be expanded in the later parts of the manuscript. What are the most important results of this study? What should readers take away? How does what you report connect to flavonoid bioactivity? Also, no statistical analysis has been done on any of the appropriate data. The work described is fascinating overall and should be published, but in an understandable way. In short, the text needs an overhaul, and statistical comparisons need to be conducted.
Please see specific comments and suggestions below:
Introduction: Please give examples of plant species where the tested flavonoids are found. Quercetin is very common, but this would add interest. Clarify your research question/hypothesis in lines 106-113, perhaps by combining this section together. Highlight the most important part of your study so this serves as an anchor later in the manuscript.
- Line 114- please spell out FRIT-ATR.
Results/Discussion: The most serious concern here is that no statistics are used for any of the data. Please do statistical analysis on the data as appropriate. Not all of your data needs it, but you include DPPH and other comparisons that need analysis.
- Typically, assays like DPPH are run on samples with some sort of control like Trolox or vitamin c along with a vehicle negative control for comparison. I understand that quercetin is often used as a positive, but please justify why you don’t include controls or please include them.
- Figure 3A MCE at 1200 t/s, should the black dot be red?
- Section 3.2 is very diffuse and I could not understand the point. Lines 350-278, how do these relate with your data? Please tie in your discussion with figures as this is a combined section.
- What exactly do you want readers to notice about your MS data? Please label figures with important products or absence of them, and discuss differences in the text and why they are important.
- Figure 6, much better explained. But what does this have to do with your experimental questions? Just an example?
- Lines 446-447- how do the antioxidant activity correspond with your MS data? Lines 447-448, please expand or point to specific figure.
- Figure 7- again, at least a negative control or blank should be run along with the flavonols. Even vehicle will show minute lipid peroxidation as a normal biological process. If not, please explain.
- Lines 454-498 are very diffuse, they refer to a table that perhaps should be included in the main text with this section shortened. Lines 499-515, how does this relate to your data? Lines 520-537, again too much discussion for a supplemental figure.
- Figure 8, what are you trying to say? And, how do figure 8 and 9 relate to your flavonols?
- Consider including table S5 in the manuscript body to help quantify and explain Figure 11.
- Figure 12- as you’re not doing correlation analysis here, remove dotted lines. They imply a linear relationship, and the flavonols don’t show this- interesting and worthy of discussion. If you want to anchor the reader’s eye, consider a line between dots instead that would highlight their true pattern.
- Lines 677-698, reference relevant figure throughout.
- 721-743- shorten dramatically please.
Conclusion: This section lists your experiments. You already described them, so please tie in your results to the larger implications about these compounds in general, as laid out in your introduction. How do the physical properties you find between flavonols and lipids prevent or alleviate oxidative damage from a chemistry perspective?
Author Response
Dear Reviewer,
thank you for your comments and suggestions. We have revised our manuscript and implemented them. In the attachment, please find the revised manuscript with all the changes already implemented (for easier reading). The specified line numbers in the answers to your comments correspond to the version with implemented changes.
Thank you again for consideration of our revised manuscript.
Best regards,
Suzana Šegota, PhD, Division of Physical Chemistry
Ruđer Bošković Institute
Bijenička cesta 54,
HR-10 000 Zagreb, Croatia
E-mail: ssegota@irb.hr
Reviewer #2: General Summary: This study has a very interesting premise. The authors set out to determine the physical and biochemical characteristics of lipid membranes that are potentially protected from oxidative damage by quercetin, myricetin, and myricitrin, flavonols with established antioxidant and other bioactivity. They use very sophisticated chemistry instrumentation and methods to answer their questions. However, after a relatively strong Introduction section, the manuscript is almost impossible to decipher. The Results/Discussion section is very explanation-heavy but does not connect important concepts back to the study’s results. The authors need to focus on and highlight their data more, rather than extensive descriptions that do not relate to flavonols. Also, the data needs to be made much more understandable. The Methods section is very descriptive, but these details don’t need to be expanded in the later parts of the manuscript. What are the most important results of this study? What should readers take away? How does what you report connect to flavonoid bioactivity? Also, no statistical analysis has been done on any of the appropriate data. The work described is fascinating overall and should be published, but in an understandable way. In short, the text needs an overhaul, and statistical comparisons need to be conducted.
Answer: Thank you for the detailed feedback. The descriptive parts of the manuscript were shortened and the main results were highlighted. Introduction was modified to highlight the main idea of the work. The main emphasis is on the ability of flavonols to suppress lipid peroxidation. This type of bioactivity has been examined using several techniques, and the most important findings were summarized in the discussion of each paragraph, as well as in the conclusion. Appropriate statistical analysis was included where possible. Specific answers and explanations are below.
Please see specific comments and suggestions below:
Question: Introduction: Please give examples of plant species where the tested flavonoids are found. Quercetin is very common, but this would add interest. Clarify your research question/hypothesis in lines 106-113, perhaps by combining this section together. Highlight the most important part of your study so this serves as an anchor later in the manuscript.
Answer: As suggested, we made corrections in the manuscript. Examples of plant species were given where the tested flavonols are found (lines 100-103). Introduction was modified in order to clarify our hypothesis and we pointed out the most important part of our study (lines 115-138).
Question: Line 114- please spell out FRIT-ATR.
Answer: As suggested, we made corrections in the manuscript (line 128).
Question: Results/Discussion: The most serious concern here is that no statistics are used for any of the data. Please do statistical analysis on the data as appropriate. Not all of your data needs it, but you include DPPH and other comparisons that need analysis.
Typically, assays like DPPH are run on samples with some sort of control like Trolox or vitamin c along with a vehicle negative control for comparison. I understand that quercetin is often used as a positive, but please justify why you don’t include controls or please include them.
Answer: Positive control, e.g. ascorbate, could be used to express antioxidative activity in ascorbate equivalents. However, the main goal of our paper was to compare the protective properties of flavonols and to correlate them with their structure and positioning inside the membrane. Therefore, we found it sufficient to only use the DPPH assay of flavonols to compare their antioxidative effect in the solution with the one in the membrane. Furthermore, the usage of ascorbate as positive control has inherent problems. Namely, it is known that the ascorbate radicals at higher concentrations can undergo recombination and dismutation. This process can cause a non-linearity in the intensity-concentration curve and make it difficult to interpret which species cause a decrease in DPPH signal, making the method unreliable. Furthermore, the kinetics of this process is highly dependent on the temperature, pH, presence of light, metal ions, etc. Due to the complexity of the problem, this is the subject of our further study – the detailed study of other factors affecting the reactions, as well as the controls and samples that must be further considered.
Statistical analysis (section 2.9) was performed where possible (antioxidative assay of flavonols in the solution and the extent of lipid peroxidation by FTIR).
Question: Figure 3A MCE at 1200 t/s, should the black dot be red?
Answer: The black dot was corrected.
Question: Section 3.2 is very diffuse and I could not understand the point. Lines 350-278, how do these relate with your data? Please tie in your discussion with figures as this is a combined section. What exactly do you want readers to notice about your MS data? Please label figures with important products or the absence of them and discuss differences in the text and why they are important.
Answer: Our MS data concerns LP products (section 3.2), and the differences that occur when flavonols are incorporated within the liposomes. Accordingly, section 3.2 was dramatically shortened. We removed unnecessary part in order to make text more concise. Also, the part describing the derivatization was moved to the experimental section (section 2.4., lines 227-234) and data interpretation procedure was additionally clarified. Since MS spectra changed during the observed time frame, it was not possible to show all products in Figures 4 and 5. However, three products were selected as an example and added to the figures which additionally clarify the disappearance of the products with the addition of flavonols. The main findings of these experiments were pointed out based on your suggestions.
Question: Figure 6, much better explained. But what does this have to do with your experimental questions? Just an example?
Answer: Figure 6 is used as an example of DOPC spectrum before lipid peroxidation. Additionally, we have pointed out two bands in the figure that were used in the analysis to tie it in with the results presented in the paper.
Question: Lines 446-447- how do the antioxidant activity correspond with your MS data? Lines 447-448, please expand or point to a specific figure.
Answer: As suggested, we further explained the idea of these measurements and discussed the results.
Question: Figure 7- again, at least a negative control or blank should be run along with the flavonols. Even vehicle will show minute lipid peroxidation as a normal biological process. If not, please explain.
Answer: Since the object of our measurement was the extent of lipid peroxidation in the systems with added flavonoids, the role of the control sample is played by DOPC without inserted flavonoids (spectrum shown in Fig 6). The value r0 corresponds to the relative change in absorbance of the DOPC sample without flavonoids after lipid peroxidation, which was calculated relative to the absorbance of the DOPC sample before lipid peroxidation (without flavonols). In that case, r0 corresponds to the control sample and is included in the calculation of Ri (by the definition, R0 = 0, where the subscript i = 0 denotes the control sample). In our case, the only important parameter is the change in the intensity of carbonyl peak (which corresponds to the extent of peroxidation), regardless of the way that the reaction was initiated (by air or Fenton reaction).
Question: Lines 454-498 are very diffuse, they refer to a table that perhaps should be included in the main text with this section shortened. Lines 499-515, how does this relate to your data? Lines 520-537, again too much discussion for a supplemental figure.
Answer: As suggested, table S3 was included in the main text (table 1) and the text was shortened. Results obtained from DLS/ELS measurement (section 3.4.1) were shortened, summarized and clarified.
Question: Figure 8, what are you trying to say? And, how do figure 8 and 9 relate to your flavonols?
Answer: As suggested, we further explained Figures 8 and 9 in the manuscript (section 3.4.2).
Question: Consider including table S5 in the manuscript body to help quantify and explain Figure 11.
Answer: As suggested, we included table S5 in the manuscript (Table 2).
Question: Figure 12- as you’re not doing correlation analysis here, remove dotted lines. They imply a linear relationship, and the flavonols don’t show this- interesting and worthy of discussion. If you want to anchor the reader’s eye, consider a line between dots instead that would highlight their true pattern.
Answer: As suggested, dotted lines were removed.
Question: Lines 677-698, reference relevant figure throughout.
Answer: As suggested, relevant figures were referenced throughout the paragraph.
Question: 721-743- shorten dramatically please.
Answer: As suggested, the paragraph was shortened.
Question: Conclusion: This section lists your experiments. You already described them, so please tie in your results to the larger implications about these compounds in general, as laid out in your introduction. How do the physical properties you find between flavonols and lipids prevent or alleviate oxidative damage from a chemistry perspective?
Answer: As suggested, the conclusion was rewritten with the most important results clarified and further explained.

Reviewer 3 Report
Dear Authors, Dear Editor,
I was asking to review the manuscript « The structural integrity of the model lipid membrane during induced lipid peroxidation: the role of flavonols in the inhibition of lipid peroxidation » by Sadžak et al.
In their study the authors studied the structural integrity of model lipid membrane during peroxidation and the protective role of flavonols. They used in that purpose multi-technique approach and analysed the lipid bilayers in term of roughness, thickness, elasticity and fluidity. They also used other techniques suc as Mass Spectrometry (MS) and Fourier Transform Infrared Spectroscopy (FTIR to identify oxidation products and follow the reaction. Finally, atomic Force Microscopy (AFM), Force Spectroscopy (FS), Small Angle X-Ray Scattering (SAXS), and Electron Paramagnetic Resonance (EPR) experiments were also used to study the consequence of the peroxidation on the model membrane and the protective effect of flavolnols. Authors conclusions were that their multi-technique approach is able to follow the change caused by peroxidation in models of lipid membranes and to study the protective effect of antioxidants such as flavonols. The authors also concluded that their approach provides insight into the pathophysiology of cellular oxidative injury.
Although I am not an expert in the techniques used in this study, I found this study very interesting and convincing. I will have some comments to be addressed.
Line 92 : « …consequently, play the role in the rate of the lipid peroxidation. » = In my opinion ad based on the conclusions of the authors the insertion of flavonols into membrane does not play a role in lipid peroxidation but rather inhibits it, so please change this sentence.
Figure 3 : Please provide a bigger version with a improved resolution as well for the figure as it is hard to see the titles of the axis
Lines 410-411 : « Additionally, acrolein vanishes in all samples except one (x=0.1 MCE) with incorporated flavonols, which is of great biological significance. » Please explain here why inhibition of acrolein formation has a great biological significance.
Line 442 : « formed LPPs significantly.. However » There is twice the dot. Please remove one.
Lines 445-448 : In seems, based on Fig 7 and contrarly to what the authors stated lines 446-447 : (« These results are in accordance with the products obtained using MS spectrometry. »), that the three flavonols gave results different from the previous assays. Indeed in Fig 7 QUE and MCI were the more active followed by MCE whereas MCI and MCE were more active than QUE in the previous assays. Did the authors have an explanation for that ? Could they comment this difference of results on the efficiency of the three flavonols to inhibit peroxidation depending of the assay in the text ?
Line 491 : « contrary, the shift towards more positive zeta potential » It seems that there is a double spacement between the word « towards » and « more ».
Line 504 : I think for homogeneity, it is better to keep the same naming of flavonols in the Results part and use MCI or MCE here as well.
Lines 652-653 : Same (please check the rest of the Results part and figures’s legend as well).
Line 681 : « myricitrin within the DOPC bilayer takes place more towards » It seems that there is a double spacement between the word « bilayer » and « takes ».
Line 793 : « DLS measurements showed a decrease of » Please correct to « decrease in ». Please check all the text for « decrease/increase OF » and change to « decrease/increase IN » (i.e. Line 802).
From a more physiological point of view, did the authors (or other studies) tried to do the same analysis but with liposomes containing cholesterol ? Cholesterol in naturally present in the biological eukaryote membrane and is able to influence the membrane rigidity/fluidity by it-self. It will be interesting to see if incorporing cholesterol to DOPC liposomes, using physiological ratio of Cholesterol and phospholipids may change something in the results obtained by the authors.
Regards.
Author Response
Dear Reviewer,
thank you for your comments and suggestions. We have revised our manuscript and implemented them. In the attachment, please find the revised manuscript with all the changes already implemented (for easier reading). The specified line numbers in the answers to your comments correspond to the version with implemented changes.
Thank you again for consideration of our revised manuscript.
Best regards,
Suzana Šegota, PhD, Division of Physical Chemistry
Ruđer Bošković Institute
Bijenička cesta 54,
HR-10 000 Zagreb, Croatia
E-mail: ssegota@irb.hr
Reviewer #3: In their study the authors studied the structural integrity of model lipid membrane during peroxidation and the protective role of flavonols. They used in that purpose multi-technique approach and analysed the lipid bilayers in term of roughness, thickness, elasticity and fluidity. They also used other techniques suc as Mass Spectrometry (MS) and Fourier Transform Infrared Spectroscopy (FTIR to identify oxidation products and follow the reaction. Finally, atomic Force Microscopy (AFM), Force Spectroscopy (FS), Small Angle X-Ray Scattering (SAXS), and Electron Paramagnetic Resonance (EPR) experiments were also used to study the consequence of the peroxidation on the model membrane and the protective effect of flavolnols. Authors conclusions were that their multi-technique approach is able to follow the change caused by peroxidation in models of lipid membranes and to study the protective effect of antioxidants such as flavonols. The authors also concluded that their approach provides insight into the pathophysiology of cellular oxidative injury.
Although I am not an expert in the techniques used in this study, I found this study very interesting and convincing. I will have some comments to be addressed.
Question: Line 92 : « …consequently, play the role in the rate of the lipid peroxidation. » = In my opinion ad based on the conclusions of the authors the insertion of flavonols into membrane does not play a role in lipid peroxidation but rather inhibits it, so please change this sentence.
Answer: As suggested, we made corrections in the manuscript (line 93).
Question: Figure 3 : Please provide a bigger version with a improved resolution as well for the figure as it is hard to see the titles of the axis
Answer: As suggested, we inserted a figure with improved resolution.
Question: Lines 410-411 : « Additionally, acrolein vanishes in all samples except one (x=0.1 MCE) with incorporated flavonols, which is of great biological significance. » Please explain here why inhibition of acrolein formation has a great biological significance.
Answer: The importance of acrolein disappearance was explained (lines 404-406), with the relevant referance added.
Question: Line 442 : « formed LPPs significantly.. However » There is twice the dot. Please remove one.
Answer: As suggested, we made corrections in the manuscript.
Question: Lines 445-448 : In seems, based on Fig 7 and contrarly to what the authors stated lines 446-447 : (« These results are in accordance with the products obtained using MS spectrometry. »), that the three flavonols gave results different from the previous assays. Indeed in Fig 7 QUE and MCI were the more active followed by MCE whereas MCI and MCE were more active than QUE in the previous assays. Did the authors have an explanation for that ? Could they comment this difference of results on the efficiency of the three flavonols to inhibit peroxidation depending of the assay in the text ?
Answer: Since Ri values were calculated from the obtained integral absorbances, their SE is significant. Therefore, we did not do the comparison of the flavonols based on their antioxidative activity, since the obtained differences are within the error interval. We performed statistical analysis to ensure that the direct comparison is not possible in this case and it showed that the differences between flavonoids and their molar fractions are not significant. Although the direct comparison could not be made, the results that are in accordance with the MS results correspond to the inhibitory activity which all flavonols showed at all molar fractions (explained in lines 479-483).
Question: Line 491 : « contrary, the shift towards more positive zeta potential » It seems that there is a double spacement between the word « towards » and « more ».
Answer: As suggested, we made corrections in the manuscript.
Question: Line 504 : I think for homogeneity, it is better to keep the same naming of flavonols in the Results part and use MCI or MCE here as well. Lines 652-653 : Same (please check the rest of the Results part and figures’s legend as well).
Answer: As suggested, we made corrections in these sections.
Question: Line 681 : « myricitrin within the DOPC bilayer takes place more towards » It seems that there is a double spacement between the word « bilayer » and « takes ».
Answer: As suggested, we made corrections in the manuscript.
Question: Line 793 : « DLS measurements showed a decrease of » Please correct to « decrease in ». Please check all the text for « decrease/increase OF » and change to « decrease/increase IN » (i.e. Line 802).
Answer: As suggested, we made corrections in the manuscript.
Question: From a more physiological point of view, did the authors (or other studies) tried to do the same analysis but with liposomes containing cholesterol ? Cholesterol in naturally present in the biological eukaryote membrane and is able to influence the membrane rigidity/fluidity by it-self. It will be interesting to see if incorporing cholesterol to DOPC liposomes, using physiological ratio of Cholesterol and phospholipids may change something in the results obtained by the authors.
Answer: We are aware that cholesterol is naturally present in biological membranes. However, in this paper, we focused on model membranes consisting only of lipid molecules to identify structural changes upon insertion of flavonoids and lipid peroxidation and correlate them with changes of packing of phospholipid chains. Since cholesterol additionally alters these properties, we are planning on including cholesterol in model membranes in our future experiments, which would mimic physiological conditions even better. We believe that the research presented in this paper will help us to enhance our understanding of effects occurring in more complex systems which will include additional parameters.

Reviewer 4 Report
The paper is well-written and clear. The English is clear and concise. The manuscript employs mass spectrometry, Fourier Transform Infrared Spectroscopy, EPR, atomic force microscopy, Force spectrography, and SAXS to analyze lipid bilayer properties with and without quercetin, myricetin, and myricitrin followed by H2O2/labile iron-mediated oxidative stress. Lipid peroxidation, membrane roughness, thickness, elasticity, and fluidity were examined. The flavonoids exerted differing, but significant membrane protective properties. The methodology is sound and the experiment, and their controls are well-designed. My only concern is that 1 mM H2O2 is a high concentration, however the authors never claimed this dose was physiologic and it would be needed for the studies preformed.
The paper contributes to the field of membrane-flavonoid interactions and the protective effects of flavonoids on membranes following oxidative stress. Many of the experiments are unique and extremely interesting. The paper should be published.
Author Response
Dear Reviewer,
thank you for your comments and suggestions. We have revised our manuscript and implemented them. In the attachment, please find the revised manuscript with all the changes already implemented (for easier reading). The specified line numbers in the answers to your comments correspond to the version with implemented changes.
Thank you again for consideration of our revised manuscript.
Best regards,
Suzana Šegota, PhD, Division of Physical Chemistry
Ruđer Bošković Institute
Bijenička cesta 54,
HR-10 000 Zagreb, Croatia
E-mail: ssegota@irb.hr
Reviewer #4: Comments and Suggestions for Authors
The paper is well-written and clear. The English is clear and concise. The manuscript employs mass spectrometry, Fourier Transform Infrared Spectroscopy, EPR, atomic force microscopy, Force spectrography, and SAXS to analyze lipid bilayer properties with and without quercetin, myricetin, and myricitrin followed by H2O2/labile iron-mediated oxidative stress. Lipid peroxidation, membrane roughness, thickness, elasticity, and fluidity were examined. The flavonoids exerted differing, but significant membrane protective properties. The methodology is sound and the experiment, and their controls are well-designed.
Question: My only concern is that 1 mM H2O2 is a high concentration, however the authors never claimed this dose was physiologic and it would be needed for the studies preformed.
Answer: As requested, the explanation regarding hydrogen peroxide concentration was added with the relevant reference (lines 163-169).
The paper contributes to the field of membrane-flavonoid interactions and the protective effects of flavonoids on membranes following oxidative stress. Many of the experiments are unique and extremely interesting. The paper should be published.

Round 2
Reviewer 1 Report
The authors answered almost all the questions of the referee and carried out the required changes . This significantly improved the manuscript that is now suitable for publication